# *Agrobacterium* expressing a type III secretion system delivers *Pseudomonas* effectors into plant cells to enhance transformation

Vidhyavathi Raman [1,4], Clemencia M. Rojas[1,5], Balaji Vasudevan[1], Kevin Dunning[1], Jaydeep Kolape [1], Sunhee Oh[1], Jianfei Yun[1], Lishan Yang[1], Guangming Li[1], Bikram D. Pant[1,2,3], Qingzhen Jiang[1] & Kirankumar S. Mysore [1,2,3 ✉]

*Agrobacterium*-mediated plant transformation (AMT) is the basis of modern-day plant bio-technology. One major drawback of this technology is the recalcitrance of many plant species/varieties to *Agrobacterium* infection, most likely caused by elicitation of plant defense responses. Here, we develop a strategy to increase AMT by engineering *Agrobacterium tumefaciens* to express a type III secretion system (T3SS) from *Pseudomonas syringae* and individually deliver the *P. syringae* effectors AvrPto, AvrPtoB, or HopAO1 to suppress host defense responses. Using the engineered *Agrobacterium*, we demonstrate increase in AMT of wheat, alfalfa and switchgrass by ~250%–400%. We also show that engineered *A. tumefaciens* expressing a T3SS can deliver a plant protein, histone H2A-1, to enhance AMT. This strategy is of great significance to both basic research and agricultural biotechnology for transient and stable transformation of recalcitrant plant species/varieties and to deliver proteins into plant cells in a non-transgenic manner.

[1] Noble Research Institute, LLC, Ardmore, OK 73401, USA. [2] Institute for Agricultural Biosciences, Oklahoma State University, Ardmore, OK 73401, USA. [3] Department of Biochemistry and Molecular Biology, Oklahoma State University, Stillwater, OK 74078, USA. [4] Present address: Department of Plant and Microbial Biology, University of Minnesota, St. Paul, MN 55108, USA. [5] Present address: Department of Entomology and Plant Pathology, University of Arkansas, Fayetteville, AR 72701, USA. ✉email: kmysore@okstate.edu

The soil-borne phytopathogen *Agrobacterium tumefaciens* causes crown gall tumors on various dicotyledonous plants by transferring a piece of its DNA (transfer DNA; T-DNA) and virulence proteins into host cells through the type IV secretion system (T4SS)[1]. The ability to transfer T-DNA from *Agrobacterium* to plants has been widely adopted to generate transgenic plants expressing genes of interest for research purposes or for commercial applications[2,3]. However, the generation of transgenic plants has several drawbacks, including the low transformation efficiency of some plant species/varieties. Plant defense responses against *Agrobacterium* significantly contribute to recalcitrance[4].

Active plant defense against microbial infection relies on innate immune responses triggered by several layers of microbial recognition. The first layer involves the perception of conserved microbial molecules called pathogen-associated molecular patterns (PAMPs) by pattern recognition receptors (PRRs) leading to PAMP-triggered immunity (PTI) that often is sufficient to prevent pathogen growth[5]. Perception of one of the most abundant *A. tumefaciens* proteins, the PAMP elongation factor thermo unstable (EF-TU)[6], by the Arabidopsis PRR EF-TU receptor (EFR) activates a set of signaling events and defense responses that reduces *Agrobacterium*-mediated plant transformation (AMT)[7]. Arabidopsis *efr* mutants are more susceptible to AMT[7]. Therefore, reducing or dampening plant basal immunity is not only essential for a successful pathogen to cause disease, but also will aid in AMT.

In contrast to *A. tumefaciens* which has a T4SS, many Gram-negative plant pathogenic bacteria have a type III secretion system (T3SS) to deliver bacterial proteins directly into their eukaryotic hosts. Many such delivered proteins, known as type III effectors (T3Es), have virulence functions that interfere with host cellular processes to block PTI, thus allowing bacteria to thrive in their hosts and cause disease[8]. T3SSs are macromolecular machines consisting of protein complexes that assemble a needle-like structure that spans the bacterial inner and outer membranes and traverses the plant cell wall and cell membrane[9]. The synthesis of effector proteins is co-regulated with proteins encoding the type III secretion apparatus[1]. Effector proteins contain an export signal at their N-termini with the characteristic composition of amphipathic and polar amino acids[10,11]. Although the effector protein content varies among pathogens, the genes encoding the type III secretion apparatus are broadly conserved and functional when heterologously expressed. For example, the T3SS from *Pseudomonas syringae* pv. *syringae* 61 (*Pss61*) and *Erwinia chrysanthemi* expressed in *P. fluorescens* and *Escherichia coli*, respectively, allowed these non-pathogenic bacteria to deliver bacterial proteins into plants[12,13].

Many T3Es, including AvrPto from *P. syringae* pv. *tomato*, can suppress plant basal defense[9,14]. The interaction between AvrPto and kinase domains of the PRRs Flagellin sensitive2 (FLS2) and EFR leads to the suppression of PTI[15]. When AvrPto is expressed under the control of an inducible promoter, Arabidopsis becomes more susceptible to transient AMT[16]. Transient expression of *AvrPto* by co-infiltration also improves transient transgene expression in *Brassica* sp.[17]. Earlier, we showed that Arabidopsis and *Nicotiana benthamiana* plants compromised for plant defense were more susceptible to AMT[18]. Recently, the increased transient transformation was achieved in Arabidopsis *NahG* expressing plants in which the defense signaling hormone salicylic acid is reduced[19]. Even though these results demonstrate that AMT can be increased by decreasing plant defense responses, practically this strategy cannot be used in the field because of the need for the generation of transgenic plants. An alternative approach to increase plant transformation is by altering the expression of host factors (other than genes involved in plant defense responses) that play a role in plant transformation and regeneration. Several plant proteins, including histones, have been identified to play a role in AMT[20–25]. However, altering the expression of host factors needs a transgenic approach that is time-consuming and creates additional hurdles for deregulation.

Here, we report a strategy based on engineering *A. tumefaciens* with a T3SS to deliver proteins that suppress plant defense and/or increase transformation. *P. syringae* pv. *tomato* T3Es such as AvrPto, AvrPtoB, or HopAO1, when co-delivered along with T-DNA through engineered *A. tumefaciens* during the transformation process, increase transformation efficiency in Arabidopsis, *N. benthamiana*, wheat, alfalfa, and switchgrass. Delivery of the plant protein histone H2A-1 also increases transformation efficiency.

## Results

**T3SS from *Pseudomonas syringae* pv. *syringae* 61 is functional in *A. tumefaciens*.** T3SS encoding genes cloned from *Pss61*[26], contained in the plasmid pLN18, are functional in *P. fluorescens*[27] and *E. coli*[12]. Here, we tested whether the expression of the *Pss61* T3SS in *A. tumefaciens* is functional to secrete and translocate T3Es. We introduced pLN18, containing the T3SS genes, and a plasmid that can express the effector protein AvrPto tagged with the fluorescent reporter PhiLOV into *A. tumefaciens* (Fig. 1a). To monitor *hrp*-dependent effector secretion into the medium, the *A. tumefaciens* strain expressing T3SS and *AvrPto-PhiLOV* along with appropriate control strains were cultured in *hrp*-derepressing medium[28]. Both cell pellet and supernatant fractions were used for immunoblot analysis. AvrPto-PhiLOV could be found in both the cell pellet and the supernatant fractions for *A. tumefaciens* containing pLN18 and expressing *AvrPto-PhiLOV* (Fig. 1b). An *A. tumefaciens* strain expressing *AvrPto-PhiLOV* without pLN18 showed the presence of AvrPto-PhiLOV only in the cell pellet and not in the supernatant fraction (Fig. 1b). These results demonstrate that an *A. tumefaciens* strain expressing a T3SS is able to express a T3E and secrete it from *A. tumefaciens* into the culture medium.

To demonstrate that the T3E secreted from the engineered *A. tumefaciens* strain can be delivered into plant cells, we used a previously established split GFP system[29,30] by infiltrating *N. benthamiana* leaves with *A. tumefaciens* that contain a *GFP_{1-10}* gene within the T-DNA of a binary vector (Fig. 1c), followed by infiltration of the same leaves with *A. tumefaciens* containing pLN18 and expressing *AvrPto-GFP_{11}* (Fig. 1c). Live-cell imaging showed green fluorescence signals inside the plant epidermal cells resulting from the assembly of full-length GFP from the interaction of GFP_{11} and GFP_{1-10}, indicating delivery of AvrPto-GFP_{11} into plant cells (Fig. 1d). As expected, green fluorescence was not observed in leaves infiltrated with *Agrobacterium* strains lacking either the T3SS or the tagged effector protein gene (Fig. 1d). FM4-64 staining of the leaves showed plasma membrane localization of AvrPto-GFP, similar to previous reports[29] (Supplementary Fig. 1a). Using the same split GFP system, we also showed delivery of other T3Es, including AvrPtoB or AvrB (Supplementary Fig. 1b). To validate our results further, we used another approach to directly deliver AvrPto-PhiLOV from *A. tumefaciens* containing pLN18 into plant cells (Supplementary Fig. 1c). In addition to demonstrating that the T3SS is functional in *A. tumefaciens* to secrete T3E in culture and translocate them to plant cells, assembly of a full-length GFP by independent translocation through T4SS and T3SS highlights that both T3SS and T4SS can operate in *A. tumefaciens* with T3SS translocating proteins and T4SS translocating both proteins and T-DNA.

**T3Es delivered by a T3SS in *A. tumefaciens* improves transformation.** The T3E AvrPto suppresses plant innate immunity[31] that hinders AMT[32]. Inducible expression of *AvrPto* in transgenic

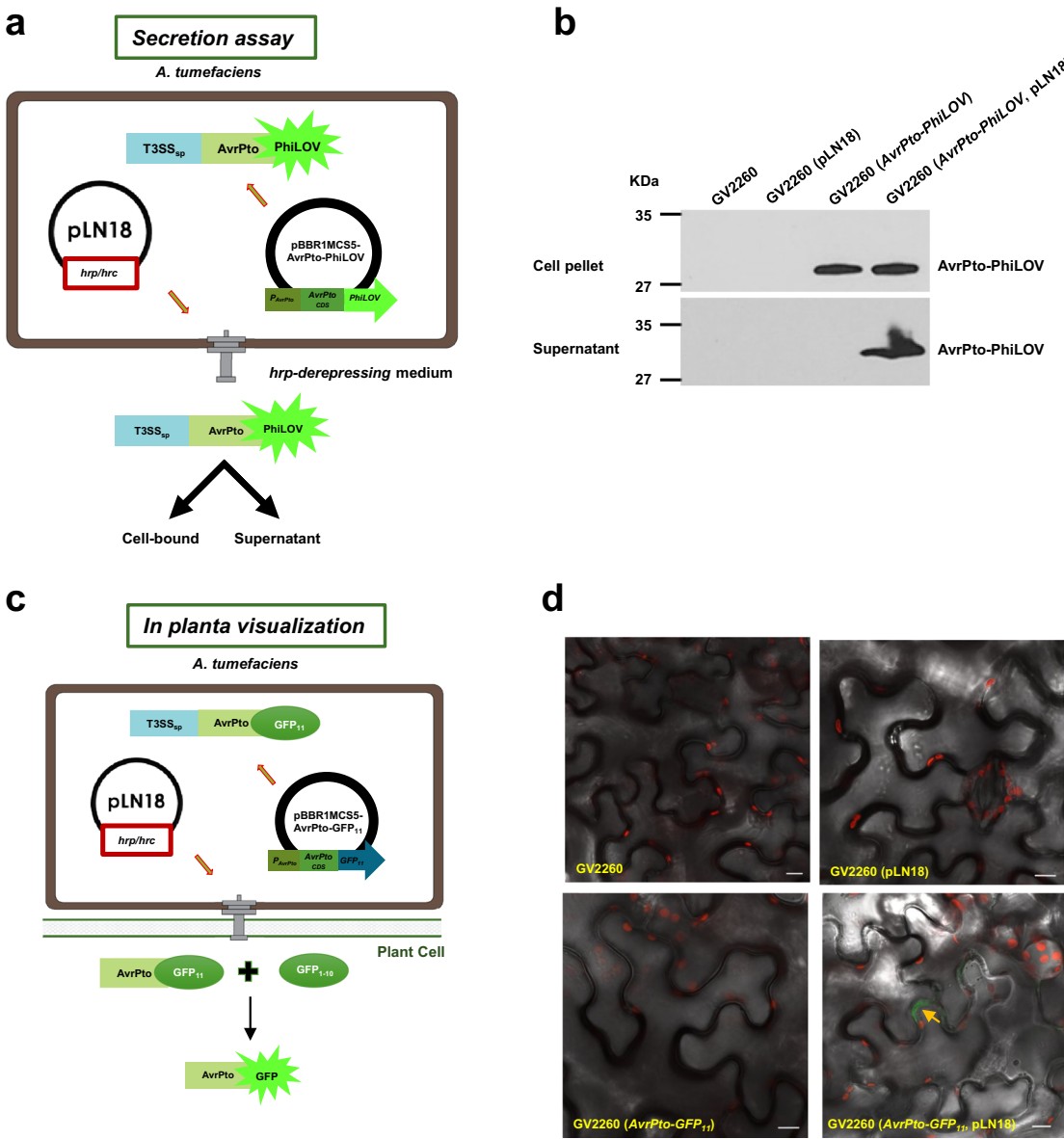

**Fig. 1 *Pseudomonas* type III secretion system when expressed in *A. tumefaciens* can deliver T3Es to plant cells. a** Schematics of engineering *A. tumefaciens* to deliver proteins through a T3SS assay. *A. tumefaciens* strain GV2260 was mobilized with plasmids containing the *P. syringae* T3SS (pLN18) and T3E *AvrPto* tagged with *PhiLOV* (pBBR1MCS5-AvrPto-PhiLOV) to express the T3SS and *AvrPto-PhiLOV*. Promoter ($P_{AvrPto}$) and coding sequences (CDS) of *AvrPto* without a stop codon were fused to codon-optimized sequences of *PhiLOV*. CDS of *AvrPto* includes sequences encoding a type III secretion signal peptide (T3SS_sp). **b** AvrPto-PhiLOV secreted in the culture medium by the engineered *A. tumefaciens* were detected by immunoblotting. GV2260 derived *A. tumefaciens* strains grown in *hrp*-derepressing medium were separated into cell pellet and supernatant fractions and probed with PhiLOV-specific antibody. **c** Schematics of engineering *A. tumefaciens* to deliver proteins through a T3SS for *in planta* visualization. *A. tumefaciens* strain GV2260 was mobilized with pLN18 and a plasmid containing *AvrPto-GFP11* to express a T3SS and AvrPto-GFP11. **d** AvrPto-GFP11 delivered through a T3SS of an engineered *A. tumefaciens* can complement GFP1-10 expressed in plants. Representative confocal images of *N. benthamiana* leaves transiently expressing GFP1-10 individually infiltrated with *A. tumefaciens* strains GV2260, GV2260 (pLN18), GV2260 (*AvrPto-GFP11*) and GV2260 (*AvrPto-GFP11*, pLN18) are shown. Confocal microscopy was used to visualize GFP fluorescence 48 h post-infiltration. GFP signals were pseudo-colored to green and chlorophyll autofluorescence is shown in red. AvrPto-GFP11 translocated to plant cells through a T3SS of engineered *A. tumefaciens* complemented GFP1-10 produced *in planta* to form functional GFP. Scale bars, 10 μm. Experiments were repeated three times with similar results. Source data are provided as a Source Data file.

Arabidopsis increases transient transformation efficiency[16]. To determine if AvrPto delivered through a T3SS can increase transient transformation, we transferred pLN18 (containing T3SS genes) and a plasmid that expresses *AvrPto* under its native promoter into the disarmed *A. tumefaciens* strain EHA105 containing a binary vector with a *β-glucuronidase* (*GUS*)-intron gene within the T-DNA (Supplementary Fig. 2a). This engineered *A. tumefaciens* strain, along with appropriate controls, was

infiltrated into the leaves of Arabidopsis plants. *GUS* expression significantly increased when *A. tumefaciens* expresses a T3SS and *AvrPto* (Fig. 2a and b). To determine if T3SS delivery of AvrPto can also increase stable transformation, we introduced pLN18 and a plasmid expressing *AvrPto* into the tumorigenic strain *A. tumefaciens* A208 (Supplementary Fig. 2b). This engineered *A. tumefaciens* strain A208 was used for Arabidopsis root transformation assay[33]. Root segments inoculated with the

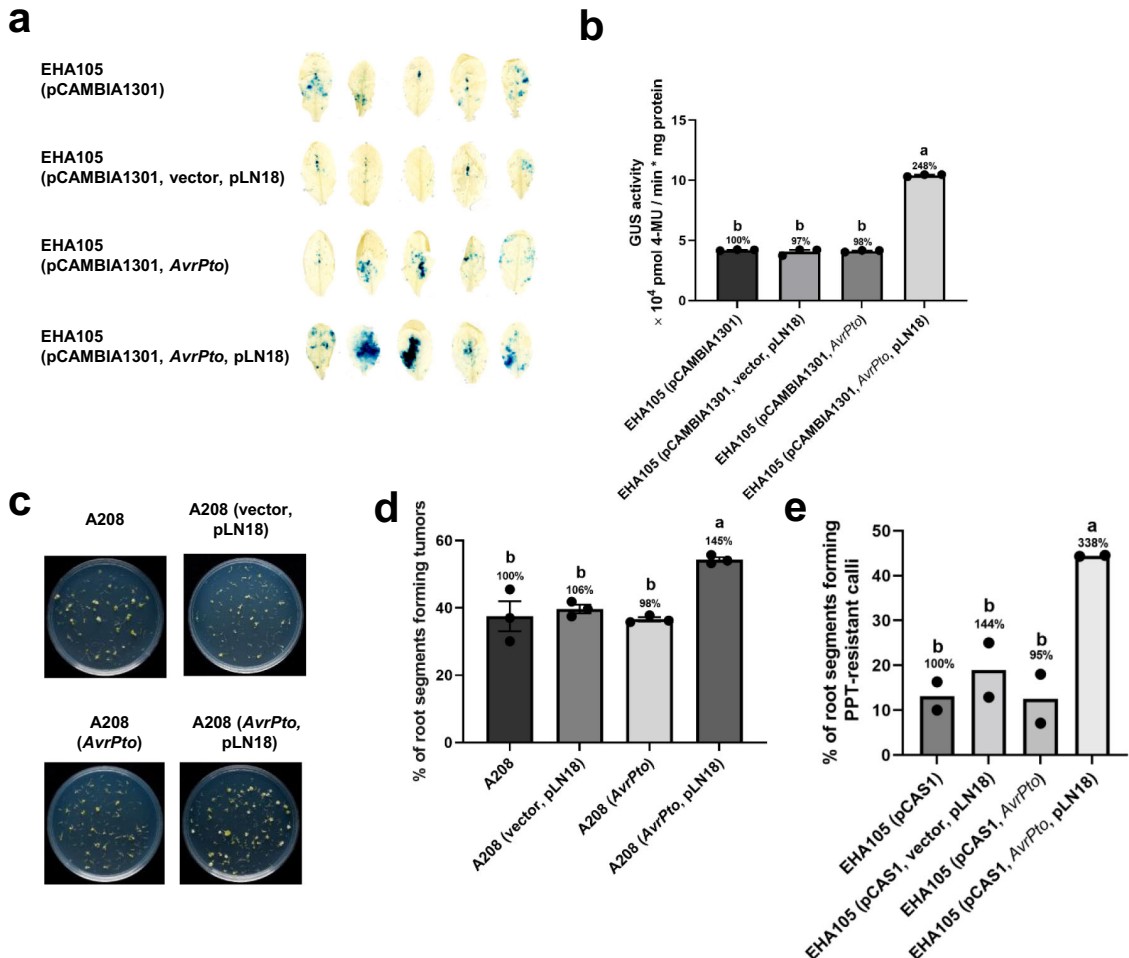

**Fig. 2 Engineered *A. tumefaciens* strains expressing a T3SS and *AvrPto* increase the transient and stable transformation efficiency of Arabidopsis.**
**a**, **b** Transient transformation assay. **a** GUS staining of Arabidopsis leaves infiltrated with *A. tumefaciens* strain EHA105 ($A_{600} = 0.2$) harboring the binary vector pCAMBIA1301 (harbors *GUS* gene within the T-DNA) alone or in combination with pLN18, an empty vector, or a plasmid expressing *AvrPto*. Four days after infiltration, the leaves were stained with X-gluc and photographed. **b** Quantitative fluorometric GUS assays with extracts from leaves of plants treated as in **a**. **c**, **d** Root tumor assay. **c** Arabidopsis root segments were infected with the tumorigenic *A. tumefaciens* strain A208 ($A_{600} = 0.01$) expressing *AvrPto* in combination with or without pLN18. *A. tumefaciens* strains A208 and A208 (vector, pLN18) were also included as negative controls. Photographs were taken 4 weeks after *A. tumefaciens* infection. **d** Root segments forming tumors were counted from the experiment in **c** and the percentage of root segments forming tumors was calculated. **e** Root callus assay. Arabidopsis root segments were infected with non-tumorigenic *A. tumefaciens* strain EHA105 harboring the binary vector pCAS1 ($A_{600} = 0.001$) expressing *AvrPto* in combination with or without pLN18. *A. tumefaciens* strains EHA105 (pCAS1) and EHA105 (pCAS1, empty vector, pLN18) were included as negative controls. Four weeks after infection, root segments forming phosphinothricin (PPT)-resistant calli were counted and the percentage of root segments forming PPT-resistant calli was calculated. Data presented in **b** and **d** are mean ± standard error of three replicates. Bars with different letters are significantly different based on Tukey's post-hoc one-way ANOVA analysis ($p < 0.05$). Brown-Forsythe test was done to test for variance ($p = 0.6830$ for **b**, 0.1433 for **d**). Data presented in **e** are mean of two replicates. Bars with different letters are significantly different based on Tukey's post-hoc two-way ANOVA analysis ($p = 0.0102$). Experiments were repeated three times with similar results. Source data are provided as a Source Data file.

*A. tumefaciens* strain carrying pLN18 and expressing *AvrPto* developed significantly more tumors compared to controls (Fig. 2c and d). T-DNA encoded *iaaM*, *iaaH*, and *ipt* genes of tumorigenic strains cause overproduction of phytohormones such as auxin and cytokinin in plants that lead to tumor formation[34] and may affect plant defense responses[35]. Therefore, we tested stable transformation efficiency in Arabidopsis roots using non-tumorigenic strain EHA105 carrying binary vector pCAS1[20] that gives phosphinothricin (PPT) resistant calli because of a chimeric *nos-bar* gene expression in plants. Consistent with the tumor results, engineered *A. tumefaciens* strain carrying pLN18 and expressing *AvrPto* developed significantly more PPT-resistant calli compared to controls (Fig. 2e and Supplementary Fig. 3a). In addition, we also tested if our engineered *A. tumefaciens* strain can also enhance the floral dip transformation that is commonly

used in Arabidopsis. We used a low concentration of *A. tumefaciens* ($A_{600} = 0.1$) to see subtle differences between the *A. tumefaciens* strains used. Surprisingly, delivery of AvrPto through T3SS increased the floral dip transformation efficiency by two-fold (Supplementary Fig. 3b).

Similar experiments were performed in a different plant species, *N. benthamiana*, using the disarmed strain *A. tumefaciens* GV2260 for transient expression and the tumorigenic strain *A. tumefaciens* A348 for stable leaf disk transformation[22]. These results were similar to those using Arabidopsis wherein expression of a T3SS and *AvrPto* in *A. tumefaciens* significantly increased both transient and stable transformation (Fig. 3a–d).

Like AvrPto, several other T3Es have the ability to suppress plant basal defense to establish/aid the growth of pathogens and

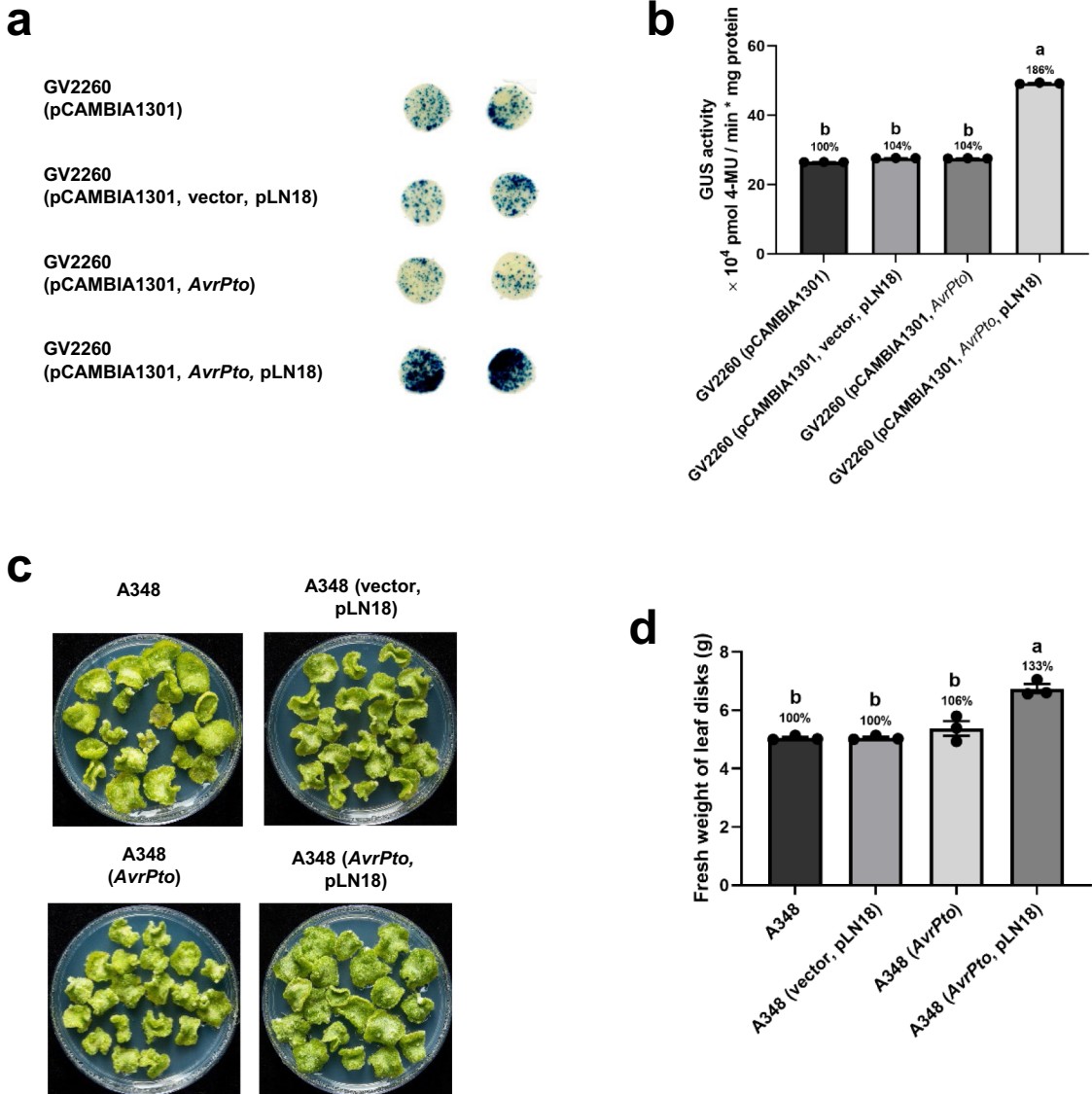

**Fig. 3 Engineered *A. tumefaciens* strains expressing a T3SS and *AvrPto* increase the transient and stable transformation efficiency of *N. benthamiana*.**
**a, b** Transient transformation assay. **a** GUS staining of *N. benthamiana* leaf disks infiltrated with *A. tumefaciens* strain GV2260 ($A_{600} = 0.0005$) harboring pCAMBIA1301 and expressing *AvrPto* in combination with or without pLN18. *A. tumefaciens* strains GV2260 (pCAMBIA1301) and GV2260 (pCAMBIA1301, vector, pLN18) were also included as controls. Four days after infiltration, the leaf disks were stained with X-gluc and photographed. **b** Quantitative fluorometric GUS assays with extracts from leaves of plants treated as in **a**. **c, d** Leaf disk transformation assay. **c** *N. benthamiana* leaf disks were infected with the tumorigenic *A. tumefaciens* strain A348 ($A_{600} = 0.1$) expressing *AvrPto* with and without pLN18. *A. tumefaciens* strains A348 and A348 (vector, pLN18) were also included as negative controls. Photographs were taken 10 days after *A. tumefaciens* infection. **d** Fresh weight of leaf disks was evaluated from the experiment in **c**. Data presented in **b** and **d** are mean ± standard error of three replicates. Bars with different letters are significantly different based on Tukey's post-hoc one-way ANOVA analysis ($p < 0.05$). Brown-Forsythe test was done to test for variance ($p = 0.2380$ for **b** and 0.3645 for **d**). Experiments were repeated three times with similar results. Source data are provided as a Source Data file.

cause disease[14]. To further examine the effect of other T3Es on AMT, we selected two T3Es from *P. syringae* pv. *tomato*: *AvrPtoB* and *HopAO1*. Similar to AvrPto, both AvrPtoB and HopAO1 significantly increased the percentage of root segments forming tumors and the weight of leaf disk tumors (Fig. 4 and Supplementary Fig. 4). As a negative control, another set of Arabidopsis root tumor assays was carried out using a *HopAI1* construct. Since *HopAI1* is targeting the PTI pathway by inhibiting MAPKs downstream of PAMP receptors[36], we hypothesized that expression *HopAI1* would not increase the susceptibility of the host to *A. tumefaciens* infection. As expected, we did not see any increase in transformation efficiency in the negative control (Supplementary Fig. 5a). These results suggest that T3Es when delivered through T3SS of engineered *A. tumefaciens* can enhance both transient and stable transformation in *N. benthamiana* and Arabidopsis.

**Delivery of plant defense suppressing T3Es improves the transformation of crop plants.** Both Arabidopsis and *N. benthamiana* are highly susceptible to stable AMT, and therefore the increase in transformation efficiency we observed by co-delivery of T3Es was only incremental for these species. Despite continuous efforts by many groups, efficient and reproducible *Agrobacterium*-mediated wheat transformation remains challenging[37,38]. Most reports of AMT of wheat have

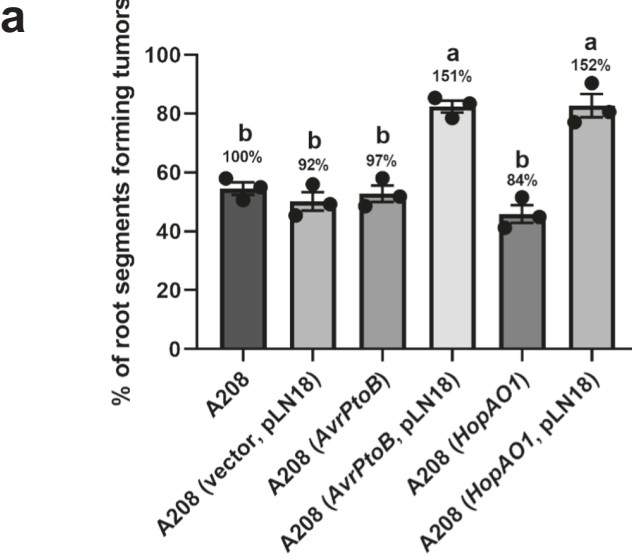

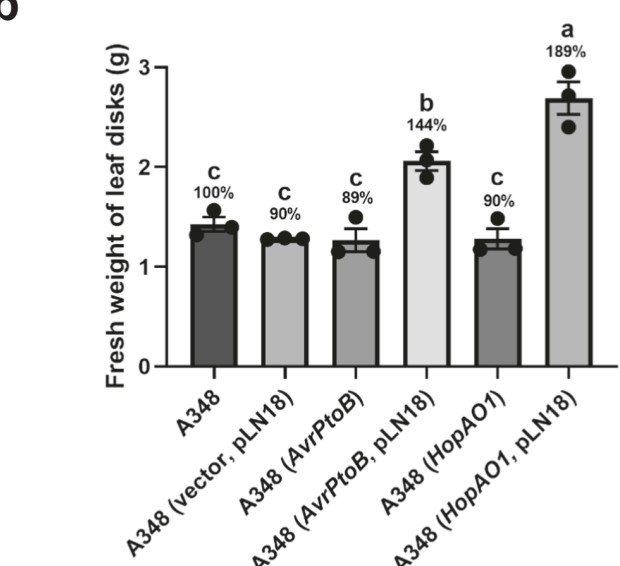

**Fig. 4 Engineered *A. tumefaciens* strains expressing a T3SS and *AvrPtoB* or *HopAO1* greatly increase the stable transformation efficiency of Arabidopsis and *N. benthamiana*. a** Root tumor assay. Arabidopsis root segments were infected with *A. tumefaciens* strain A208 ($A_{600}$ = 0.1) expressing the T3Es *AvrPtoB* or *HopAO1* in combination with or without pLN18. *A. tumefaciens* strains A208 and A208 (vector, pLN18) were included as negative controls. Root segments forming tumors were counted 4 weeks after *A. tumefaciens* infection and the percentage of root segments forming tumors was calculated. **b** Leaf disk transformation assay. *N. benthamiana* leaf disks were infected with tumorigenic *A. tumefaciens* strain A348 ($A_{600}$ = 0.01) expressing *AvrPtoB* or *HopAO1* in combination with or without pLN18. *A. tumefaciens* strains A348 and A348 (vector, pLN18) were included as negative controls. Fresh weight of leaf disks was evaluated 10 days after *A. tumefaciens* infection. **a, b** Percentage of root segments forming tumors and fresh weight of leaf disks were subjected to statistical analysis. Data presented are mean ± standard error of three replicates. Bars with different letters are significantly different based on Tukey's post-hoc one-way ANOVA analysis (one-sided) ($p < 0.05$). Brown-Forsythe test was done to test for variance ($p = 0.9723$ for **a** and 0.7349 for **b**). Experiments were repeated three times with similar results. Source data are provided as a Source Data file.

focused on the model spring wheat genotypes Fielder and Bobwhite[39]. We chose to determine if AMT of the wheat cultivar Fielder could be improved by co-delivery of T3Es. Immature embryos of wheat were infected with engineered *A. tumefaciens* strain AGL1 (pANIC6B) harboring pLN18 and a plasmid expressing T3E. The number of immature embryos that produced transgenic calli and subsequently regenerated shoots were counted. Transgenic plants derived from these regenerated shoots were tested for the activity and presence of reporter genes by GUS histochemical staining and PCR analysis of the *GUSPlus* and *hph* genes (Supplementary Fig. 6). *A. tumefaciens* strains individually delivering AvrPto, AvrPtoB, or HopAO1 through engineered T3SS greatly increased the percentage of individual transgenic plants obtained (Fig. 5a and Supplementary Fig. 7). The *A. tumefaciens* strain expressing AvrPto produced the best results, with a transformation efficiency ~400% that of the control strain lacking the T3SS components. These results indicate that *A. tumefaciens* with an engineered T3SS that delivers T3Es can increase the transformation efficiency of a recalcitrant crop species. Wheat transformation assay was also carried out using *A. tumefaciens* strains expressing *HopAI1*. Delivery of HopAI1 through T3SS did not have any effect on wheat transformation similar to the results obtained for Arabidopsis root assay (Supplementary Fig. 5b).

To determine if the engineered *A. tumefaciens* strains can also be used to improve the transformation efficiency of other commercial crop plants, we used our engineered strain that can deliver AvrPto through T3SS on alfalfa line R2336 and switchgrass line NFCX01. We observed 260% increase in transformation efficiency in alfalfa and 400% increase in transformation efficiency in switchgrass (Fig. 5b and c). These results indicate that engineered *A. tumefaciens* delivering *AvrPto* can be used to enhance AMT in many commercially important crop plants.

**Virulence gene expression is not altered in the engineered *A. tumefaciens* strains expressing T3SS.** As shown above, *A. tumefaciens* strains expressing T3SS and T3Es effectively increased AMT efficiency. The expression of T3Es in *A. tumefaciens* may increase virulence gene (*vir*) expression and thus increase transformation efficiency. To test this, we measured the expression of several *vir* genes in engineered *A. tumefaciens* A208 strains using reverse transcription-quantitative PCR (RT-qPCR). No major differences were observed in *virA*, *virB2*, *virD2*, and *virE3* gene induction, after acetosyringone treatment, among *A. tumefaciens* strains with or without the T3SS + T3E (Supplementary Fig. 8). These results, along with those reported in Fig. 2, indicate that the increase in transformation by *A. tumefaciens* strains expressing T3SS and T3Es is not due to increased expression of *vir* genes and is most likely due to the delivery of T3Es into plant cells.

**AvrPto delivered through engineered *A. tumefaciens* T3SS reduces the expression of plant defense genes.** Based on the role of AvrPto in suppressing plant defense[15], and our results showing the delivery of AvrPto along with T-DNA into plants increases AMT (Figs. 2 and 3), we speculated that the increase in AMT efficiency was due to the suppression of plant defense responses. To show that AvrPto delivered through engineered *A. tumefaciens* T3SS can suppress plant defense responses, we infected Arabidopsis roots with tumorigenic *A. tumefaciens* A208 expressing a T3SS and *AvrPto*, or with negative controls, and measured the expression of well-known PTI marker genes including *FLG22-induced receptor-like kinase 1* (*FRK1*), and

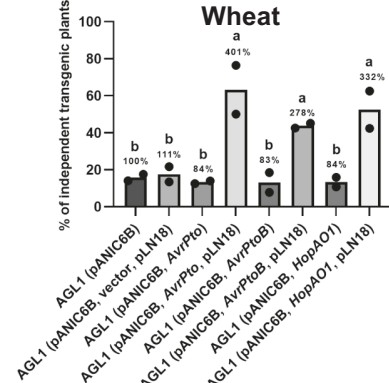

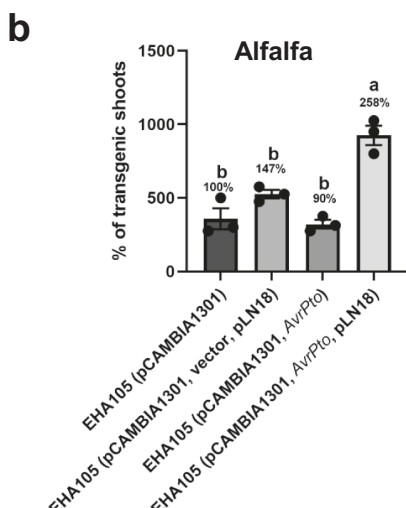

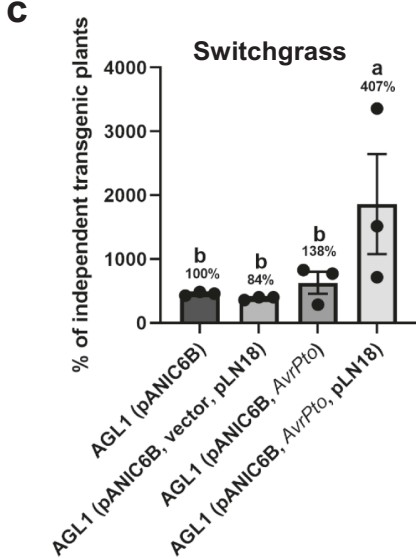

**Fig. 5 Engineered *A. tumefaciens* strains expressing a T3SS and T3Es greatly increase the stable transformation efficiency of wheat, alfalfa, and switchgrass. a** Wheat transformation. Immature embryos of wheat were infected with *A. tumefaciens* strain AGL1 harboring the binary vector pANIC6B ($A_{600} = 0.5$) and expressing T3Es *AvrPto*, *AvrPtoB* or *HopAO1* in combination with or without pLN18. *A. tumefaciens* AGL1 (pANIC6B) alone and AGL1 (pANIC6B) containing pLN18 and vector were included as controls. Data presented are the mean percentage of independent transgenic plants obtained from two independent experiments ($n = 2$ experiments; number of immature embryos used per experiment: 28–90). Bars with different letters are significantly different based on Tukey's post-hoc two-way ANOVA analysis ($p = 0.0003$). **b** Alfalfa transformation. Leaflets from alfalfa line R2336 were infected with *A. tumefaciens* strain EHA105 harboring the binary vector pANIC6B ($A_{600} = 0.12$) and expressing *AvrPto* in combination with or without pLN18. *A. tumefaciens* AGL1 (pANIC6B) alone and AGL1 (pANIC6B) containing pLN18 and vector were included as controls. Data presented are the percentage of transgenic shoots obtained per leaflet ($n = 3$ replicates; number of leaflets used per replicate = 8) shown as mean ± standard error. Bars with different letters are significantly different based on Tukey's post-hoc one-way ANOVA analysis (one-sided) ($p < 0.05$). Brown-Forsythe test was done to test for variance ($p = 0.7841$). Experiments were repeated two times with similar results. **c** Switchgrass transformation. Calli induced from switchgrass NFCX01 inflorescences were infected with *A. tumefaciens* strain AGL1 harboring the binary vector pANIC6B ($A_{600} = 0.22$) and expressing *AvrPto* in combination with or without pLN18. *A. tumefaciens* AGL1 (pANIC6B) alone and AGL1 (pANIC6B) containing pLN18 and vector were included as controls. Data presented are the percentage of independent transgenic plants obtained from each callus ($n = 3$ replicates; number of calli used per replicate = 7) shown as mean ± standard error. Bars with different letters are significantly different based on Tukey's post-hoc one-way ANOVA analysis (one-sided) ($p < 0.05$). Brown-Forsythe test was done to test for variance ($p = 0.1344$). Experiments were repeated two times with similar results. Source data are provided as a Source Data file.

expressing the T3SS and *AvrPto* compared to control strains (Fig. 6). These results indicate that AvrPto, when delivered through an engineered T3SS of *A. tumefaciens*, can suppress the plant defense response, thus contributing to increased AMT efficiency.

**Delivery of a plant protein from an engineered *A. tumefaciens* strain expressing a T3SS enhances stable transformation.** AMT is a complex process involving functions of both bacterial virulence proteins and plant proteins[40]. Histone H2A-1 (encoded by the gene *HTA1*) is involved in T-DNA integration[24], and over-expression of *HTA1* and truncated *HTA1* (*tHTA1*; coding only the first 39 amino acids) in plants increases transformation efficiency[24,41,42]. To verify if plant proteins that enhance AMT can also be delivered through an engineered T3SS of *A. tumefaciens*, we chose *HTA1* and *tHTA1*. Two different promoters, along with N-terminal sequences containing a type III signal from the T3Es AvrRpm1 and AvrRps4, designated as *AvrRpm1N* and *AvrRps4N*, respectively, were selected to drive the expression of *HTA1* and *tHTA1* and export from *A. tumefaciens*. Using the engineered *A. tumefaciens* strains, we conducted Arabidopsis root and *N. benthamiana* leaf disk tumor assays. Both *HTA1* and *tHTA1* expressing *A. tumefaciens* strains enhanced the stable transformation efficiency of Arabidopsis and *N. benthamiana* (Fig. 7a–c and Supplementary Fig. 9a). Transformation assays in crop plants using engineered strains expressing *HTA1* also showed increased transformation efficiency in wheat, alfalfa, and switchgrass (Fig. 7d–f, Supplementary Fig. 9b and c). These

*NDR1/HIN1-like 10* (NHL10). Irrespective of the *A. tumefaciens* strain used, both the tested defense-related genes were induced in response to *A. tumefaciens* infection at 2 h after infection when compared to a mock-infected control (Fig. 6). However, 16 h after *A. tumefaciens* infection, transcripts of defense-related genes were significantly reduced in root samples infected with *A. tumefaciens*

## a

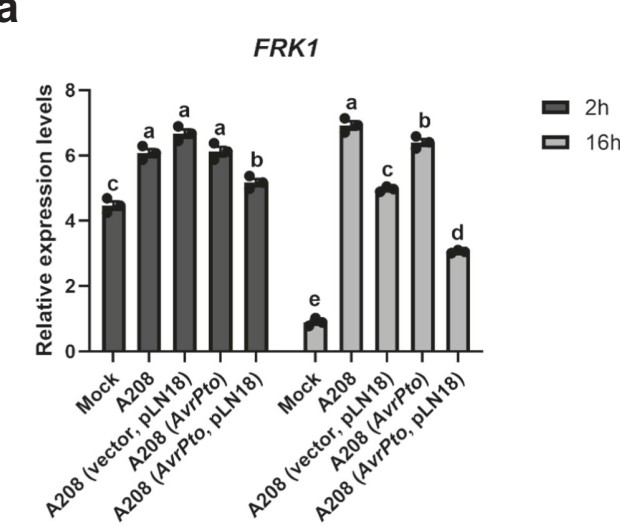

## b

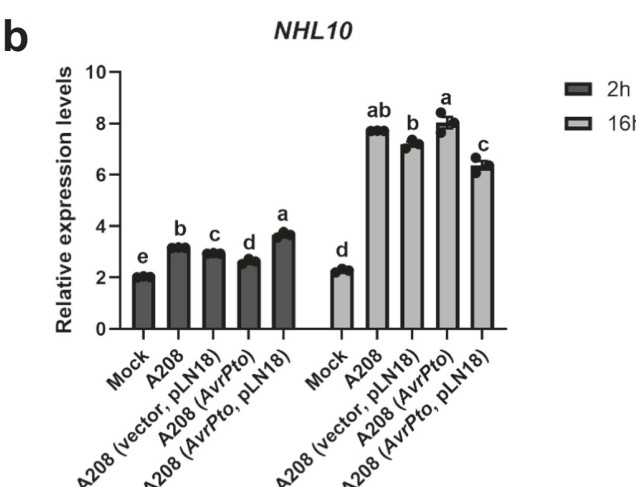

**Fig. 6 Engineered *A. tumefaciens* strains expressing a T3SS and *AvrPto* reduce expression of Arabidopsis defense genes. a, b** Relative expression levels of PTI marker genes. Roots of Arabidopsis plants were infected with tumorigenic *A. tumefaciens* strain A208 ($A_{600} = 1.0$) expressing *AvrPto* in combination with and without pLN18. Mock infection (ABM-MS medium), A208, and A208 (vector, pLN18) were included as negative controls. Expression levels of **a** *FRK1*, and **b** *NHL10* were determined using $2^{-\Delta\Delta CT}$ method with the *UBQ10* gene as a housekeeping control. Data presented are mean ± standard error of three replicates. Bars with different letters are significantly different based on Tukey's post-hoc one-way ANOVA analysis (one-sided) ($p < 0.05$). Brown-Forsythe test was done to test for variance ($p = 0.9965$ for *FRK1* 2 h, 0.4600 for *FRK1* 16 h, 0.1414 for *NHL10* 2 h, 0.2042 for *NHL10* 16 h). Experiments were repeated three times with similar results. Source data are provided as a Source Data file.

results indicate that engineered *A. tumefaciens* expressing a T3SS can also be used to deliver plant proteins to enhance AMT.

## Discussion

*A. tumefaciens* is a plant pathogen that causes crown gall disease in many plant species and has been widely used in the field of plant biotechnology and functional genomic studies, including the recently developed genome-editing technologies[3,43]. The

application of AMT is limited to particular plant species/varieties due to low transformation efficiency, mainly caused by the induction of plant defense responses[7,44–46]. In the present work, we expressed the *Pss61* T3SS in *A. tumefaciens* to deliver T3Es or a plant protein to Arabidopsis, *N. benthamiana*, and wheat to enhance AMT. We also showed that the increased transformation by T3E AvrPto is associated with an attenuated plant defense response. Many plant pathogenic bacteria contain T3SSs that act as an essential virulence determinant to deliver T3Es that remodel normal plant cellular functions, compromising the plants' immune responses and promoting bacterial multiplication[47]. We used the T3SS of *P. syringae* species because it is well characterized[48,49].

The T3SS has some limitations in the size and structure of proteins that it can deliver to the host cell. For example, the fusion of T3Es with a large fluorescent protein-like GFP interferes with effector secretion and translocation because of the inability of the type III ATPase to unfold the tightly packed GFP domain[50]. Therefore, we explored the possibility of using a small 13 KDa fluorescent reporter PhiLOV[51], and successfully demonstrated its use in visualizing effector expression, secretion, and translocation to plant cells (Fig. 1 and Supplementary Fig. 1). Previously, PhiLOV has been used to tag T3Es of animal pathogens to monitor effector secretion and translocation[52] and to visualize translocation and localization of *A. tumefaciens* virulence proteins[53]. In another study, a split GFP system was used to monitor the secretion of T3Es of *P. syringae* and *R. solanacearum*[29,30]. Recently, a T3SS gene cluster from *Xanthomonas euvesicatoria*, which is known to secrete large effectors[54] was shown to secrete effectors into the plants[55]. However, heterologous expression of a *Xanthomonas* T3SS has not yet been reported[55].

A previous attempt by Tsuda et al.[16] to express T3Es other than AvrPto *in planta* to increase the transient transformation of Arabidopsis was not successful, likely because these effectors are not targeting the PTI pathway at the PAMP receptor level. We therefore selected the *P. syringae* T3Es AvrPtoB and HopAO1, which are involved in the early immune signaling of PTI. These proteins target PRRs and disrupt their functions[14,56]. In our study, both AvrPtoB and HopAO1, when co-delivered with T-DNA, increased the stable transformation of Arabidopsis, *N. benthamiana*, and wheat (Figs. 4 and 5a).

Research in several laboratories had previously suggested that AMT of wheat was not reproducible across laboratories[37,38]. Previously, an ~5% transformation efficiency was reported[38]. Recently, up to 25% AMT frequency of "Fielder" was achieved by manipulating various parameters, including the stage of the donor material and pretreatment by centrifugation[38]. However, using the conventional transformation method, we could only achieve ~15% stable transformation efficiency (Fig. 5a). With our strategy of co-delivering T3Es along with the T-DNA, we could achieve up to 63% transformation efficiency by delivering AvrPto through a T3SS (Fig. 5a). High transformation efficiency is critical for gene-editing technology, especially in agronomically important plants like wheat. The limitation of this strategy is that it cannot be used in plant varieties that are recalcitrant to regeneration.

To understand whether increased AMT observed in our study is due to higher bacterial virulence or suppression of plant immunity, we investigated the expression of *Agrobacterium vir* genes and plant defense genes after *Agrobacterium* infection. Constitutive expression of *A. tumefaciens virA* can increase tumor formation on Arabidopsis[57]. Increased transformation efficiencies were also achieved by additional copies of heterologous *virB* and *virG* genes when compatible Ti plasmids were used[58,59]. In the present study, the expression of *vir* genes in the *Agrobacterium*

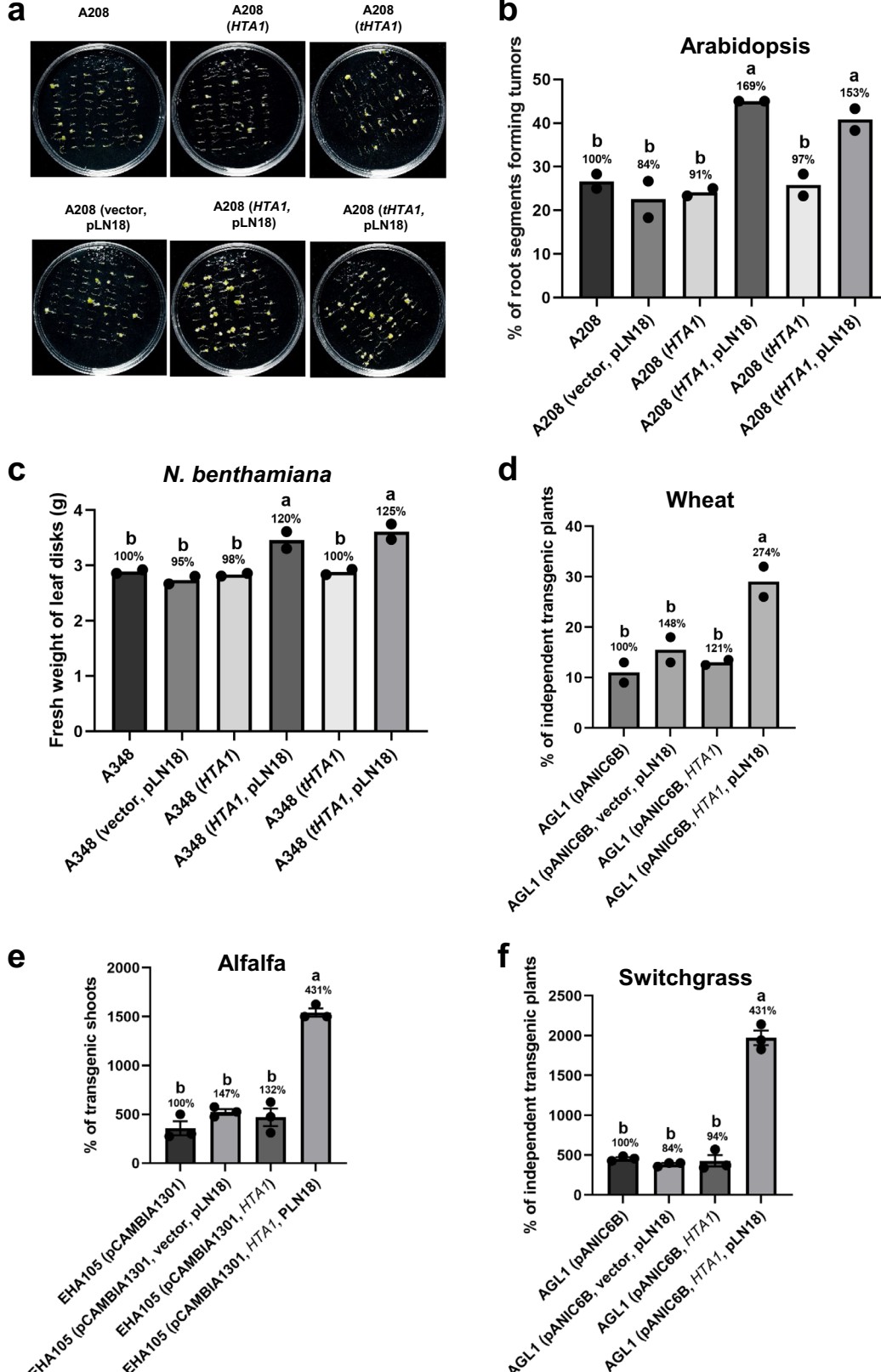

strain expressing T3SS and *AvrPto* did not show major differences from control strains. *Agrobacterium* infection induces the expression of various plant defense genes, including *PR* genes and those encoding chitinases and defensins[7,32,60]. *FRK1*, a commonly used molecular marker gene to study T3E-mediated suppression of plant defense[61,62], is activated by PAMPs and not by other stress-related signals[62]. Many studies designed to optimize media composition for AMT rely on the expression of *FRK1* and *NHL10* to monitor plant defense responses[63–65]. We showed that the expression of *FRK1*, and *NHL10* is reduced by an engineered

**Fig. 7 Engineered *A. tumefaciens* strains expressing a T3SS and the plant histone *HTA1* increase the stable transformation efficiency. a, b** Root tumor assay. **a** Arabidopsis root segments were infected with *A. tumefaciens* strain A208 ($A_{600} = 0.01$) expressing *HTA1* or truncated *HTA1 (tHTA1)* under the control of the *AvrRpm1N* (promoter + type III signal) in combination with or without pLN18. *A. tumefaciens* strains A208 and A208 (vector, pLN18) were included as negative controls. Photographs were taken 4 weeks after *A. tumefaciens* infection. **b** Root segments forming tumors were counted from the experiment in **a** and the percentage of root segments forming tumors was calculated. **c** Leaf disk transformation assay. *N. benthamiana* leaf disks were infected with tumorigenic *A. tumefaciens* strain A348 ($A_{600} = 0.01$) expressing *HTA1* or *tHTA1* under the control of the *AvrRps4N* (promoter + type III signal) in combination with or without pLN18. *A. tumefaciens* strains A348 and A348 (vector, pLN18) were included as negative controls. Fresh weight of leaf disks was evaluated 10 days after *A. tumefaciens* infection. **b, c** The percentage of root segments forming tumors and the fresh weight of leaf disks were subjected to statistical analysis. Data presented are mean of two replicates. Bars with different letters are significantly different based on Tukey's post-hoc two-way ANOVA analysis ($p = 0.0034$ for **b** and 0.0055 for **c**). Experiments were repeated three times with similar results. **d** Stable transformation efficiency in wheat. Immature embryos of wheat were infected with AGL1 (pANIC6B) carrying different plasmids as described in Fig. 5. Expression of *HTA1* was driven by *AvrRps4N*. Data presented are the percentage of independent transgenic plants from two independent experiments as mean. Bars with different letters are significantly different based on Tukey's post-hoc two-way ANOVA analysis ($p = 0.0375$). **e, f** Alfalfa and switchgrass transformation as described in Fig. 5b and c for *A. tumefaciens* expressing *HTA1*. Bars with different letters are significantly different based on Tukey's post-hoc one-way ANOVA analysis (one-sided) ($p < 0.05$). Brown-Forsythe test was done to test for variance (alfalfa $p = 0.7011$; switchgrass $p = 0.4290$). Source data are provided as a Source Data file.

*Agrobacterium* strain that delivers AvrPto through a T3SS, thus increasing AMT (Fig. 6).

One way to increase the efficiency of AMT is by suppressing the plant defense response as described above or by altering the expression of plant genes that play a role in AMT[25]. To determine if the engineered *Agrobacterium* with a T3SS can deliver plant proteins into plants cells, we used *A. tumefaciens* codon-optimized sequences of *HTA1* and *tHTA1*. The expression of functional proteins in heterologous hosts is often enhanced by codon optimization[66,67]. Delivery of either HTA1 or tHTA1 through the T3SS increased AMT (Fig. 7).

Transformation and regeneration are major bottlenecks in the generation of transgenic plants or genome-edited plants. Our study aimed to improve transformation in the delivery and integration of T-DNA steps. The chimeric protein of plant developmental gene encoding GROWTH-REGULATING FACTOR 4 (GRF4) and its cofactor GRF-INTERACTING FACTOR 1 (GIF1) (GRF4-GIF1) and GRF5 were shown to enhance regeneration and transformation of both monocot and dicot species[68,69]. Specific morphogenic genes that are known to induce somatic embryogenesis and regeneration are also used to improve transformation efficiency in monocots[70,71]. Since constitutive expression of these genes has negative phenotypic and reproductive effects, altruistic transformation is used in maize and sorghum which uses the transient expression of *Baby boom* and *Wuschel2* to promote somatic embryogenesis and regeneration in nearby transformed cells[72,73]. In the future, these proteins can be potentially delivered through *Agrobacterium* strain expressing T3SS to improve somatic embryogenesis and plant regeneration.

Heterologous protein expression in plants is achieved by stable or transient transformation by delivery of genes of interest harbored in *Agrobacterium* T-DNA[74]. AMT has emerged as a vehicle for the application of clustered regularly interspaced short palindromic repeats (CRISPR)/Cas9-mediated genome editing of plants[75]. Genome-editing reagents are delivered to plants mostly through a stable transformation that requires segregation of *Cas9* by Mendelian segregation to achieve transgene-free genome-edited plants (null segregants)[43]. In addition, off-target mutations may be increased by constitutive *Cas9* expression[76], which can be significantly reduced by conditional and transient expression of *Cas9*[75]. Our approach of delivering proteins from *Agrobacterium* through a T3SS can not only increase AMT but also has the potential to alleviate the above-mentioned problems of CRISPR-mediated genome editing by delivering bacterially expressed Cas9 to plants through a T3SS instead of generating *Cas9* expressing transgenic plants. In addition, our technology side-steps the disadvantages of making transgenic plants that overexpress genes

that enhance transformation. Recently, direct delivery of proteins using *Agrobacterium's* T4SS is getting attention, particularly in the field of DNA-free genome editing. For example, Cas9 translocation to plants through the T4SS by fusion with VirF translocation signal has been shown[77]. Previously, DNA modifying proteins such as site-specific recombinase Cre[78] and homing endonuclease I-SceI[79] have been translocated to plants through the T4SS. Our technology would be a good alternative to deliver genome-editing proteins because the T3SS is evolved for fast and efficient translocation of multiple effectors[80].

## Methods

**Bacterial strains and growth conditions.** Bacterial strains and plasmids used in this study are shown in Supplementary Data 1. *E. coli* DH5α was used for molecular cloning and was grown at 37 °C in Luria-Bertani (LB) medium. HB101 was used for maintaining the helper plasmid pRK2013. *A. tumefaciens* strains were grown at 28 °C on YEP agar plates or in YEP liquid medium, *Agrobacterium* minimal medium containing sucrose (AB-sucrose)[81], *hrp*-derepressing liquid medium (HDM)[28], or mannitol glutamate/lysogeny (MG/L) medium. Acetosyringone (200 μM) added to AB-MES[65] (17.2 mM $K_2HPO_4$, 8.3 mM $NaH_2PO_4$, 18.7 mM $NH_4Cl$, 2 mM KCl, 1.25 mM $MgSO_4$, 100 μM $CaCl_2$, 10 μM $FeSO_4$, 50 mM MES, 2% glucose (w/v), pH 5.5) and ABM-MS[65] (½ AB-MES, ¼ MS, 0.25% sucrose (w/v), pH 5.5) medium was also used. Antibiotics were spectinomycin (25 μg mL$^{-1}$), carbenicillin (10 μg mL$^{-1}$), rifampicin (10 μg mL$^{-1}$), kanamycin (50 μg mL$^{-1}$), gentamycin (25 μg mL$^{-1}$), and tetracycline (5 μg mL$^{-1}$).

**Bacterial genetic manipulations and plasmid construction.** Promoter sequences of *AvrPto* (116 bp upstream of ATG) followed by coding sequences without a stop codon, from *P. syringae* pv. *tomato* strain DC3000, as well as codon-optimized *PhiLOV*2.1 sequences, were synthesized and cloned into the broad host range vector pBBR1MCS5 at Eco53kI and KpnI sites to generate pBBR1MCS5-AvrPto-PhiLOV. *AvrPto* containing its native promoter was also synthesized and cloned into the Eco53kI and KpnI site of pBBR1MCS5 to generate pBBR1MCS5-AvrPto. The coding sequences of *AvrPtoB* and *HopAO1*, along with their native promoters (from *P. syringae* pv. *tomato* strain DC3000), were synthesized and cloned into the Eco53kI and KpnI site of pBBR1MCS5. Promoter sequences from *AvrPtoB* and *HopAO1* are 93 bp and 86 bp, respectively, upstream of ATG. *AvrRpm1N* consists of 199 bp upstream of ATG and the first 267 bp of the CDS from *P. syringae* pv. *maculicola*[27,82]. *AvrRps4N* is defined as 129 bp upstream of ATG and the first 411 bp of the CDS from *P. syringae* pv. *pisi*[83,84]. Full-length and truncated *HTA1* (coding first 39 amino acids[41]) sequences codon-optimized for *A. tumefaciens* driven by promoters and N-terminal sequences of *AvrRpm1* as well as *AvrRps4* were synthesized and cloned into Eco53kI and KpnI site of pBBR1MCS5. GenScript (Piscataway, NJ) carried out all DNA syntheses reported here. *E. coli* DH5α competent cells were transformed by a standard heat-shock procedure. Electroporation was used to introduce all the plasmids into *A. tumefaciens* strains, with the exception of pLN18. Triparental mating[85] was used to mobilize pLN18 into *A. tumefaciens* strains.

**Plant materials and growth conditions.** For root transformation assays, *Arabidopsis thaliana* Columbia-0 (Col-0) plants were grown in B5 medium in an environment-controlled growth chamber at 24 °C, 70% humidity, and a 16/8 h light/dark photoperiod with the light intensity of 50–100 μE m$^{-2}$ s$^{-1}$. For leaf infiltration assays, Arabidopsis (Col-0) plants were grown in metromix soil in a controlled growth chamber at 22 °C, 75% humidity, and a 8/16 h light/dark

photoperiod with the light intensity of 140 µE m$^{-2}$ s$^{-1}$ using fluorescent tubes. *N. benthamiana* plants were grown in soil in a controlled growth room at 24 °C, 75% humidity, and a 16/8 h light/dark photoperiod with the light intensity of 140 µE m$^{-2}$ s$^{-1}$ using fluorescent tubes.

**Secretion assays and immunoblotting**. To monitor the *hrp*-dependent effector secretion into the medium, *A. tumefaciens* strain GV2260 expressing different combinations of *AvrPto-PhiLOV*, and T3SS were grown on YEP agar plates at 28 °C for 2 days. Two colonies were transferred to HDM medium and cultured for 16 h at 28 °C with shaking at 220 rpm. Pellets from bacterial cultures at $A_{600} = 0.25$ were taken for cell pellet fraction analysis. Approximately 20 mL of the cultures (normalized based on the $A_{600}$ of the bacterial cultures to get an equal protein amount) were separated into pellet and supernatant fractions by centrifuging the culture at $3220 \times g$ for 15 min at 21 °C. The top 15 mL of the supernatant solution was carefully removed without disturbing the pellet. The supernatant solution was passed through a 0.45 µm Durapore PVDF Membrane Millipore filter (Catalog No. SE1M003M00, EMD Millipore Corporation, Burlington, MA), and further centrifugation steps were carried out at 4 °C. The supernatant fractions were concentrated by ultrafiltration using Amicon Ultra-15 Centrifugal Filters (Catalog No. UFC901024, Merck Millipore Ltd, Tullgreen, Ireland), and further concentrated to ~30 µL using Amicon Ultra-0.5 Centrifugal Filters (Catalog No. UFC501024, Merck Millipore Ltd, Tullgreen, Ireland) according to the manufacturer's instructions. Proteins from the pellet as well as supernatant fractions were subjected to electrophoresis through a SDS-PAGE gel, and immunoblot analysis was carried out using PhiLOV-specific antibody (dilution 1:5000).

**Confocal microscopy**. *A. tumefaciens* strain GV2260 carrying *GFP$_{1-10}$* in a binary vector and the engineered *A. tumefaciens* strain GV2260 expressing a T3SS and various effectors tagged with *GFP$_{11}$* were grown on YEP plates containing the appropriate antibiotics for two days at 28 °C. Single colonies were inoculated into YEP liquid medium and incubated for 16 h, cells were resuspended in an induction medium containing 10 mM MES and 200 µM acetosyringone, and incubated at room temperature for 3 h with slow shaking. The $A_{600}$ of the culture was adjusted to 0.4. Bacterial strains were syringe-infiltrated on the abaxial side of four-week-old *N. benthamiana* plants. *A. tumefaciens* strain GV2260 carrying a *GFP$_{1-10}$* gene in a binary vector was infiltrated 24 h prior to infiltration of GV2260 expressing a T3SS, and effectors, as well as negative controls. Forty-eight hours after infiltration, the infiltrated area was used for acquiring confocal micrographs with a Leica SP8 confocal microscope (Leica Microsystems, Wetzlar, Germany) using Leica Application Suite X (LAS X) software version 3.5.5.19976. GFP was excited at 488 nm and emission was gathered between 493 and 550 nm. Chloroplast autofluorescence emission was gathered between 650 and 732 nm. For FM4-64 staining, leaves were infiltrated with 25 µM FM4-64 (Catalog No. T13320, Invitrogen, Carlsbad, CA) for 1 h before microscopy. Excitations were carried out at 488 and 565 nm and emissions were gathered between 493 and 551, and 700 and 750 nm. For experiments using PhiLOV tagged effectors, excitation was carried out at 458 nm and emission was gathered between 474 and 530 nm.

**Agrobacterium-mediated transient and stable transformation assays**. Arabidopsis transient transformation assays were carried out by the previously described method[86]. In brief, *A. tumefaciens* strain EHA105 carrying different plasmids was used to infiltrate five leaves of 5-weeks-old soil-grown Arabidopsis plants. All five infiltrated leaves were harvested four days after infection for assaying GUS activity using qualitative histochemical staining and from three different plants, all five leaves were pooled for measuring GUS activity by quantitative fluorometric assays. The GUS histochemical staining assay was performed as previously described with some modifications using β-glucuronidase substrate X-gluc dissolved in DMSO[87]. In brief, plant materials were stained with X-gluc staining solution [100 mM sodium phosphate buffer (pH 7.0), 0.5 mM K$_3$Fe(CN)$_6$, 0.5 mM K$_4$Fe(CN)$_6$, 0.5 mM EDTA, 0.1% Triton X-100, and 1 mg mL$^{-1}$ X-gluc] for one day at 37 °C in the dark. GUS activity was quantified using fluorometric measurements by the conversion of 4-methylumbelliferyl-β-D-glucuronide (4-MUG) (Catalog No. M9130, Sigma–Aldrich, St. Louis, MO) to 4-methylumbelliferone (4-MU)[88]. Fluorescence measurements of the 4-MU (excitation at 365 nm, emission at 455 nm) were carried out using a Tecan Infinite® 200 Pro multimode plate reader (Tecan, Switzerland) with i-control version 11 software. GUS activity was calculated using 4-MU (Catalog No. M1381, Sigma–Aldrich, St. Louis, MO) standards. The experiments were repeated three times on different days.

Arabidopsis root transformation assays were performed by following the previously described protocol[33]. In brief, axenic root segments were infected with *A. tumefaciens* A208 or its derivatives, co-cultivated for 48 h in the dark at room temperature, and transferred to MS-basal medium supplemented with cefotaxime (200 mg L$^{-1}$) and ticarcillin (100 mg L$^{-1}$). Four weeks after infection, tumor numbers were recorded.

Arabidopsis stable transformation root callus assays were carried out as previously described[20]. In brief, axenic root segments were infected with *A. tumefaciens* EHA105 containing pCAS1[20] or its derivatives ($A_{600} = 0.001$), co-cultivated for 48 h in the dark at room temperature, and transferred to a callus induction medium supplemented with cefotaxime (200 mg L$^{-1}$), ticarcillin

(100 mg L$^{-1}$), and phosphinothricin (PPT) (10 mg L$^{-1}$). Four weeks after infection, the number of root segments forming PPT-resistant calli was recorded. Arabidopsis transformation was performed by floral dip method[89]. Experiments were carried out in two biological replicates. In each experiment, ten plants were inoculated with strain EHA105 harboring different constructs ($A_{600} = 0.1$). T0 seeds were germinated on half-strength MS media containing hygromycin (20 mg L$^{-1}$). Hygromycin-resistant plants were selected and stained for GUS activity as described above.

For *N. benthamiana* transient transformation assays, engineered *A. tumefaciens* strain GV2260 carrying different plasmids were prepared and infiltrated as described above for microscopy. Leaf disks were collected after 4 days of infection and GUS activity assays were carried out as described above. *N. benthamiana* leaf disk tumor assays were carried out as previously described[22]. In brief, leaves harvested from greenhouse-grown plants were sterilized using 8% bleach for five minutes, then washed four times with sterile distilled water. Leaf disks made using a cork borer (0.9 cm) were infected with *A. tumefaciens* strain A348 or its derivatives for 15 min followed by co-cultivation on MS-basal medium for 2 days in the dark at room temperature. Leaf disks were transferred onto MS-basal medium supplemented with cefotaxime (200 mg L$^{-1}$) and ticarcillin (100 mg L$^{-1}$). Fifteen days after transfer, the fresh weights of leaf disks were measured for the leaf disk tumorigenesis assay.

**Wheat transformation**. Wheat plants (cv. Fielder) were grown in a greenhouse with a 16/8 h light/dark photoperiod and 20–22 °C day/19–21 °C night. Immature ears were collected ~14 days after anthesis. The ears were sprayed with 70% ethanol. After removing the glume, lemma, and palea, the immature embryos (IEs) were isolated from these immature seeds under a dissecting microscope in a laminar flow hood. The IEs were centrifuged in 2 mL embryo-collection medium using a fixed-angle rotor at $17,000 \times g$ at 4 °C for 10 min. The IEs were removed from the embryo-collection medium and infected in 1 mL inoculum ($A_{600} = 0.5$) of *A. tumefaciens* strain AGL1 carrying different plasmids grown in MG/L medium with 100 µM acetosyringone. The mixture was shaken at 90 rpm at room temperature for 15 min. The infected IEs were transferred onto co-cultivation medium with the scutellum side up. Plates were sealed with microfilm tape and then vacuum infiltrated for 7 min. The plates were incubated at 24 °C in the dark for 2 days. The embryo axis was removed from the IEs before transferring onto the resting medium, then subsequently cultured in the same conditions for 5 days. The IEs were subjected to two rounds of selection by transferring onto a selection medium containing 15 mg L$^{-1}$ hygromycin for 2 weeks followed by 30 mg L$^{-1}$ hygromycin for 3 weeks. The proliferating explants were then transferred onto a shoot regeneration medium with 30 mg L$^{-1}$ hygromycin and cultured at 24 °C under illumination (16/8 h light/dark) until shoots were produced. The regenerated shoots were transferred to a root regeneration medium containing 15 mg L$^{-1}$ hygromycin under the same growth conditions. Regenerated plants were transferred to soil and sampled for GUS activity staining (as described above) and genomic DNA extraction to test for the presence of transgenes by PCR for *hph* and *GUSPlus* genes. All primers used are listed in Supplementary Data 2.

**RT-qPCR**. To study the expression levels of *A. tumefaciens vir* genes, engineered *A. tumefaciens* strain A208 grown in YEP medium overnight at 28 °C was harvested and resuspended in AB-MES[65] medium containing 200 µM acetosyringone. After incubation at room temperature with minimal shaking, bacterial cells were harvested at 24 h and stored at −80 °C for RNA extraction. RNA was extracted using a NucleoSpin RNA mini kit (Catalog No. 740955, Macherey-Nagel, Düren, Germany) according to the manufacturer's instructions, including in-column genomic DNA digestion. Superscript III Reverse Transcriptase (Catalog No. 18080085, Invitrogen) was used for the synthesis of cDNAs using 1.5 µg of RNA and random hexamers (Catalog No. N8080127, Invitrogen). Quantitative PCR (qPCR) reactions were performed using a CFX Real-time PCR system (Applied Biosystems, Foster City, CA) and a KiCqStart SYBR Green qPCR ReadyMix (MilliporeSigma, St Louis, MO). Quantitative PCR data were collected using Bio-Rad CFX Manager Version 2.1.1022.0523 software. A minimum of three technical replicates and three biological replicates per experiment were done. Relative expression values were calculated using the $2^{-\Delta\Delta CT}$ method using *recA* as a housekeeping gene control. All primers used are listed in Supplementary Data 2.

For measuring the expression of plant defense genes, Arabidopsis plants were grown vertically on B5 medium for 12 days. Overnight cultures of engineered *A. tumefaciens* strain A208 grown in YEP medium at 28 °C were harvested, resuspended in ABM-MS medium containing 200 µM acetosyringone[65], and incubated at room temperature with minimal shaking for 5 h. The $A_{600}$ of all *A. tumefaciens* cultures was adjusted to 1.0 and the bacteria were pipetted onto roots as a thin layer[90]. For mock infection, ABM-MS medium was used instead of *A. tumefaciens* cultures. Roots were harvested 2 h and 16 h after infection, rinsed with water and frozen in liquid nitrogen for RNA extraction. RNA was extracted using an RNeasy plant mini kit (Catalog No. 74904, Qiagen, Valencia, CA). Samples were digested with TURBO DNase (Catalog No. AM1907, Invitrogen) to remove genomic DNA. Reverse transcription reactions were performed with 1 µg of RNA in a 20 µL reaction using Oligo(dT)$_{12-18}$ (Catalog No. 18418012, Invitrogen) and SuperScript III Reverse Transcriptase (Invitrogen). qPCR reactions were performed using a CFX Real-time PCR system (Applied Biosystems) using KiCqStart SYBR

Green qPCR ReadyMix (Catalog No. KCQS01, MilliporeSigma). A minimum of three technical replicates and three biological replicates per experiment were done. Relative expression values were calculated using the $2^{-\Delta\Delta CT}$ method using *UBQ10* as a housekeeping control. All primers used are listed in Supplementary Data 2.

**Alfalfa transformation**. Alfalfa transformation was carried out following the previously described method[91]. Briefly, *A. tumefaciens* strain EHA105 harboring different constructs were streaked and cultured on AB agar plates containing different antibiotics at 28 °C for 2–3 days. A single colony was then cultured in AB liquid media, containing the same antibiotics, overnight till the $A_{600}$ was 0.6–0.8. The young leaflets from 4-6-week-old alfalfa line R2336 plants were sterilized with 20% commercial bleach containing a drop of Tween-20 for 10 min and then washed three times with sterilized water. These leaves were then infected with the *A. tumefaciens* suspension by resuspending the pellets in a liquid infection medium to an $A_{600}$ of 0.05 or 0.12 after centrifuging the *A. tumefaciens* liquid culture at 3500 rpm for 20 min. The infected leaves were blot dried and plated on co-cultivation medium and cultured under 24 °C in the dark for 24–30 h. These tri-foliate explants were transferred onto selection medium containing 10 mg mL$^{-1}$ hygromycin and continued growth for a total of 6–8 weeks under the same conditions. During this selection period, the explants were subcultured every 2 weeks until enough resistant calli produced. The resistant calli were then transferred onto a shoot regeneration medium with 5 mg mL$^{-1}$ hygromycin and cultured at 24 °C day /20 °C night and 16/8 h light/dark photoperiod with 150 µmol m$^{-2}$ s$^{-1}$ light. Shoots regenerated after 2–3 months of transfer were counted and checked by PCR and GUS activity staining.

**Switchgrass transformation**. Switchgrass transformation was carried out according to previously reported protocol[92]. Briefly, switchgrass NFCX01 calli induced from inflorescence were infected with *A. tumefaciens* strain AGL1 carrying different plasmids. After co-cultivation in the dark chamber at 24 °C for 3 days, infected calli were transferred onto a selection medium with 50 mg L$^{-1}$ hygro-mycin and cultured in the same conditions for 6–8 weeks. Resistant calli were then transferred onto a regeneration medium with 30 mg L$^{-1}$ hygromycin and cultured in the light chamber 16/8 h light/dark photoperiod, 24 °C day/20 °C night, and 150 µmol m$^{-2}$ s$^{-1}$ light. The regenerated shoots were transferred onto a rooting medium with 10 mg L$^{-1}$ hygromycin till plants grew big enough for screening by methods described for wheat transgenic plants.

**Statistical analysis**. GraphPad Prism version 8.0.1 was used for making graphs and ANOVA tests. R version 3.5.2 was used for all Tukey's post-hoc tests.

**Reporting summary**. Further information on research design is available in the Nature Research Reporting Summary linked to this article.

## Data availability
All relevant data supporting the key findings of this study are available within the article and its Supplementary Information files. Source data are provided with this paper.

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

## Acknowledgements
This work was supported by the National Science Foundation (grant # IOS-1725122 to K.S.M.) and the Noble Research Institute, LLC. We thank James R. Alfano (University of Nebraska-Lincoln) for providing pLN18, John M. Christie (University of Glasgow) for PhiLOV antibody, Gitta Coaker (University of California, Davis) for AvrPto-GFP$_{11}$, AvrPtoB-GFP$_{11}$ and GFP$_{1-10}$, Michael Kovach (Baldwin Wallace University) for pBBR1MCS5, and Stanton Gelvin (Purdue University) for critical reading of the manuscript and providing *A. tumefaciens* strain EHA105. We also thank communications department at Noble Research Institute for photography, graphic design, and copy-editing.

## Author contributions
K.S.M. conceptualized the hypothesis. V.R., C.M.R., B.V., Q.J., and K.S.M. conceived and designed the experiments, and did data analysis and wrote the manuscript. V.R., C.M.R., B.V., K.D., J.K., S.O., J.Y., L.Y., G.L., and B.D.P conducted the experiments.

## Competing interests
The authors declare no competing interests.
