## [Peer Review File · Nature Communications]

Agrobacterium expressing type III secretion system delivers Pseudomonas effectors into plant cells to enhance transformationReviewers' Comments:

Reviewer #1:

Remarks to the Author:

The ms by Raman et al describes the introduction of a cosmid encoding a Pseudomonas T3SS in Agrobacterium for delivery of effector proteins in plant host cells. The results show that this T3SS can transfer proteins from Agrobacterium to plants and that this can be used for enhancing Agrobacterium-mediated transformation.

Although the new research expands the tool kit for Agrobacterium I miss in the discussion a comparative analysis with the well established tools for the same purpose: expression from a (co)-delivered T-DNA, and direct delivery of proteins by Agrobacterium's own T4SS, which is already an established tool in the field for delivery of heterologous proteins.

P5 The vector expressing GFP1-10, its source and markers is missing in the plasmid list.

P6 Authors argue that their result highlights the cooperation between T3SS and T4SS. But what do they mean with cooperation. The only way to get the desired result is that the T4SS delivers one day before the T-strand to express GFP1-10 and then the next day the strain expressing the T3SS is added to deliver the effector protein, which can now be seen by GFP expression. There is separate and independent activity of these two translocation systems.

P6 Stable transformation was tested with a tumorigenic strain. Why was this chosen as the binary vector delivering GUS can also be used for transformation by selecting for the hygromycin resistance as later was done for wheat. This is the more important as tumorigenic strains deliver a T-DNA with the *ipt* and *iaa*-genes, and it is known that especially auxins influence the immune response of the plant.

P6-9 There is always considerable variation in frequencies of plant transformation, which makes it hard to test variables that make only a small difference. The authors also see this, for instance visualized in figure 2a when comparing the amount of blue staining in the different samples treated in the same way. Nevertheless, the authors still observe significant differences between different treatments and only small variation within samples as shown in the graphs in figure 2. I trust that the authors have carefully analyzed their data, but it is important that they describe exactly what they did. In the M&M section it is only said that they did 3 replicates. Were these indeed 3 independent experiments done at different moments in time with independently grown samples? And each consisting of how many plants, roots, leaves of each treatment? Five as in figure 2a?

P10 The authors mention that they see greater fluorescence in plants using GFP compared to PhiLOV similar to results in animals, but they should also refer here to Roushan et al, who already showed this in their Plant J paper from 2018.

P11 The results with wheat are interesting, and are worth to be investigated further. Recent papers have shown that reproducible high frequencies of transformation can be obtained in recalcitrant crops including wheat, not so much by preventing an immune response, but rather by stimulating cellular growth via Growth Regulating Factors (Kong et al, Front Plant Science 2020; Debernardi et al, Nature Biotech 2020. Can the effect seen by the authors be due to enhanced growth (due to down regulation of defense) rather than enhanced T-DNA delivery?

P11 and 6-9 the authors have selected and used (effector) proteins from which they expected a positive effect on transformation. To exclude that any protein delivery per se would have the same effect, it would be important to show a negative control with a similar protein from which no positive effect is expected, for instance another histone H2A or a defective effector.

P11 and throughout VirB genes etc should be without capitals

P12 the authors conclude that delivering of Cas9 through the T3SS might be an advantage over generating plants expressing Cas9. I fully agree, but the authors miss here the article by Schmitz et al, Front Plant Science 2020, where such Cas9 protein delivery was accomplished by using Agrobacterium's own T4SS. Earlier it was already shown that the T4SS can be used to deliver other nucleases such as Sce1 and ZFNs and recombination proteins such as Cre/lox into plants.

Reviewer #2:

Remarks to the Author:

Improvement of the plant transformation efficiency in recalcitrant plant species/varieties is a major issue in crop improvement applications by cis/transgenic and CRISPR technologies. This manuscript describes a work combining two known concepts: (1) Presence of type III effectors (T3Es), such as AvrPto, in the plant cell can suppress plant immune response and improve the efficiency of Agrobacterium-mediated transformation (AMT); (2) The *Pseudomonas syringae* type III secretion system (T3SS) can be functionally expressed in other Gram-negative bacteria. Since the previous applications of (1) required transgenic expression of T3E in the plant, it had a limitation in applications. The idea of combining these two in this manuscript is a good one, and thus the impact of the work mainly depends on how well this idea worked out in practice. In this light, the key results are in Fig 4 – AMT of the wheat cultivar Fielder. On average 4-fold efficiency improvement by AvrPto sounds good. However, Fielder is not AMT-recalcitrant among wheat varieties. According to the authors' discussion, this improvement is based on a transformation method, which is not a very good method to start with. Thus, the improved AMT efficiency achieved in this study is not as high as the AMT efficiency achieved by refinements of the method (ref 36). It is not known if the method described in this manuscript could further improve the efficiency from the level achieved by the refined method. Furthermore, the authors should have tried with truly AMT-recalcitrant varieties of wheat – if the method works with elite varieties with no transformation success records, it would have an impact for general audience.

Improvement of the AMT efficiency in *Arabidopsis* and *N. benthamiana* has little impact for the practical purpose: nobody uses tissue-culture-based AMT for *Arabidopsis* as the floral dip AMT method is highly efficient and much less cumbersome; *N. benthamiana* has a high efficiency to start with. Also, the improvement of the transformation efficiency is rather moderate (all less than 2-fold, except for transient transformation). Thus, the results in *Arabidopsis* and *N. benthamiana* should be considered only for the method characterization.

Suppression of immune response by pathogen effectors is the matter of timing and strength. It is likely that delivery of T3E to the plant cell is not sufficiently fast and/or not sufficiently abundant to have a strong suppression effect. This limited level of immune response suppression is clear from Fig 5.

In summary, I do not think that this work has demonstrated a sufficient impact in AMT improvement for general audience. This manuscript better fits to a more specialized journal for plant transformation. Specific points.

1. Page 2, abstract and other places. "deliver the *P. syringae* effectors AvrPto, AvrPtoB, and HopAO1" This expression gives an impression that all three effectors were delivered at once. It should be clear that it was delivered one by one but not in combination.
2. Page 4, middle. "practically this strategy cannot be used in the field because these transgenic plants will be susceptible to other pathogens". This is not a good argument because the transgene can be segregated out. The real limitation of the approach is that the plants have to be already transformed with T3E transgenes.
3. Page 6, middle. "In addition to demonstrating that the T3SS is functional in *A. tumefaciens* to secrete T3E in culture and translocate them to plant cells, assembly of a full-length GFP by independent translocation through T4SS and T3SS highlights cooperation of both systems in inter-kingdom T-DNA transfer with T3SS translocating proteins and T4SS translocating both proteins and T-DNA." This sentence is not clear. Translocated proteins are different for T3SS and T4SS. And the proteins translocated by T4SS are not engineered proteins.
4. Page 7, top. The floral dip method is the commonly used method in *Arabidopsis* in practice. For a practical purpose, the transformation efficiency by the floral dip method should be investigated.
5. Page 7, middle, Fig 2gh. Is the fresh weight of leaf disks a good measure of the weight for the tumor tissue? Why not count the number of independent transgenic plants, like Fig 4?
6. Fig 4. Instead of showing % of independent transgenic plants, it should show the number of independent transgenic plants (per certain number of plates used for the tissue culture procedure). Also, the AGL1(pANIC6B, T3E, pLN18) must be compare to AGL1(pANIC6B) in the posthoc tests. In

addition, one-way ANOVA assumes equal variance. Since the variance for high values are much larger, one-way ANOVA cannot be used directly – the authors can use Bartlett test for equal variance across the strains. Probably log-transformation works for equal variance.

7. Page 10, bottom. “we also observed a greater fluorescent signal using GFP compared to PhiLOV.” Where is the data? It must be quantitative data as it is a quantitative statement.

8. Page 11, top. “these effectors target plant immune signaling components through effector-triggered immunity (ETI) and not PAMP receptors.” This statement is wrong. There are reports that AvrRpt2 and AvrRpm1 compromise basal immunity (i.e., non-ETI).

Reviewer #3:

Remarks to the Author:

The authors have used *Agrobacterium tumefaciens* to express a type III secretion system (T3SS) from *Pseudomonas syringae* and deliver the *P. syringae* effectors AvrPto, AvrPtoB, and HopAO1 to suppress host defense responses. They also report that engineered *A. tumefaciens* expressing a T3SS can deliver histone H2A-1 protein to enhance AMT.

Using T3SS to deliver proteins into plant cells to enhance *Agrobacterium*-mediated transformation efficiencies is innovative.

However, for AvrPto, AvrPtoB, and HopAO1, they authors used the three plant species, *Arabidopsis*, *N. benthamiana* and wheat for the experiments, However, the increases in transformation efficiency of both *Arabidopsis* and *N. benthamiana* are not in significant while about 400% increase was observed for wheat. However, if only one of the three plant species tested shows significant improvement in transformation efficiency, the question is how widely applicable is their technology for other plant species.

Regarding the use of an engineered *A. tumefaciens* expressing the T3SS to deliver the histone H2A-1 protein into plant cell to enhance AMT. *Arabidopsis* and *N. benthamiana* were used for the experiment. Again, the increases in transformation efficiencies of the two plant species are insignificant.

There are also some other minor problems.

Overall, I feel that the increases in transformation efficiency presented in the manuscript are not significant enough. Also, the applicability of the described technologies in other plant species is uncertain.

Reviewers' comments:

Reviewer #1 (Remarks to the Author):

The ms by Raman et al describes the introduction of a cosmid encoding a Pseudomonas T3SS in Agrobacterium for delivery of effector proteins in plant host cells. The results show that this T3SS can transfer proteins from Agrobacterium to plants and that this can be used for enhancing Agrobacterium-mediated transformation.

Although the new research expands the tool kit for Agrobacterium I miss in the discussion a comparative analysis with the well established tools for the same purpose: expression from a (co)-delivered T-DNA, and direct delivery of proteins by Agrobacterium's own T4SS, which is already an established tool in the field for delivery of heterologous proteins.

Our response: We thank the reviewer for the suggestion. We modified the discussion as suggested.

P5 The vector expressing GFP1-10, its source and markers is missing in the plasmid list.

Our response: GFP1-10 details are present in row 14 of Supplementary table 1.

P6 Authors argue that their result highlights the cooperation between T3SS and T4SS. But what do they mean with cooperation. The only way to get the desired result is that the T4SS delivers one day before the T-strand to express GFP1-10 and then the next day the strain expressing the T3SS is added to deliver the effector protein, which can now be seen by GFP expression. There is separate and independent activity of these two translocation systems.

Our response: Here we wanted to emphasize that both T3SS and T4SS can operate in the same bacterium. GFP fluorescence observed is the result of successful complementation of GFP1-10 expressed from T-DNA delivered by T4SS and GFP11 delivered by T3SS. To our knowledge, there is no report showing the existence of functional T3SS and T4SS in the same bacterium. We modified the sentences for more clarity.

P6 Stable transformation was tested with a tumorigenic strain. Why was this chosen as the binary vector delivering GUS can also be used for transformation by selecting for the hygromycin resistance as later was done for wheat. This is the more important as tumorigenic strains deliver a T-DNA with the *ipt* and *iaa*-genes, and it is known that especially auxins influence the immune response of the plant.

Our response: We agree with the reviewer that we could have used non-tumorigenic strain and we expect to see similar results as tumorigenic strain based on our past experience. We have standardized assays in our lab by infecting Arabidopsis root segments with non-tumorigenic *Agrobacterium* strain carrying plasmid pCAS1 and counting the resultant antibiotic resistant calli to measure stable transformation efficiency. We used this assay to generate transgenic calli with the engineered *Agrobacterium* strain. As shown in Figure 2e, consistent with the tumor data, we observed ~3-fold

increase in transgenic calli produced by engineered *Agrobacterium* with T3SS when compared to controls.

P6-9 There is always considerable variation in frequencies of plant transformation, which makes it hard to test variables that make only a small difference. The authors also see this, for instance visualized in figure 2a when comparing the amount of blue staining in the different samples treated in the same way. Nevertheless, the authors still observe significant differences between different treatments and only small variation within samples as shown in the graphs in figure 2. I trust that the authors have carefully analyzed their data, but it is important that they describe exactly what they did. In the M&M section it is only said that they did 3 replicates. Were these indeed 3 independent experiments done at different moments in time with independently grown samples? And each consisting of how many plants, roots, leaves of each treatment? Five as in figure 2a?

Our response: We agree with the reviewer about the variation. To account for the variation, we infiltrated several plants and in each plant five leaves were infiltrated. All five leaves were stained for GUS and shown in Fig 2a. From three different plants, all five leaves were pooled and used for measuring GUS activity by MUG assay shown in Fig 2b. We added these details to the materials and methods.

P10 The authors mention that they see greater fluorescence in plants using GFP compared to PhiLOV similar to results in animals, but they should also refer here to Roushan et al, who already showed this in their Plant J paper from 2018.

Our response: Yes, we agree. Based on the comment from Reviewer 2, we now deleted this line.

P11 The results with wheat are interesting, and are worth to be investigated further. Recent papers have shown that reproducible high frequencies of transformation can be obtained in recalcitrant crops including wheat, not so much by preventing an immune response, but rather by stimulating cellular growth via Growth Regulating Factors (Kong et al, Front Plant Science 2020; Debernardi et al, Nature Biotech 2020. Can the effect seen by the authors be due to enhanced growth (due to down regulation of defense) rather than enhanced T-DNA delivery?

Our response: It is unlikely that the bacterial effectors will enhance plant growth. Few effectors including AvrPto have been expressed in plants using inducible promoter. In most cases, constitutive overexpression of these proteins leads to plant lethality. Therefore, it is unlikely that effectors will increase plant growth by downregulation of defense. Moreover, the downregulation of defense in our case was subtle.

P11 and 6-9 the authors have selected and used (effector) proteins from which they expected a positive effect on transformation. To exclude that any protein delivery per se would have the same effect, it would be important to show a negative control with a similar protein from which no positive effect is expected, for instance another histone H2A or a defective effector.

Our response: We thank the reviewer for the suggestion. We chose HopAI1 as a negative regulator because this effector did not have an effect on PTI pathway at the receptor level. Therefore, we believed HopAI1 construct can serve as a negative control. Tumor assays in Arabidopsis roots using *HopAI1* expressing strains did not show significant increase in transformation efficiency compared to control. Transformation efficiency in wheat also showed similar results. We have now included these results in the manuscript as Supplementary Figure 5.

P11 and throughout VirB genes etc should be without capitals

Our response: We changed this as suggested.

P12 the authors conclude that delivering of Cas9 through the T3SS might be an advantage over generating plants expressing Cas9. I fully agree, but the authors miss here the article by Schmitz et al, Front Plant Science 2020, where such Cas9 protein delivery was accomplished by using Agrobacterium's own T4SS. Earlier it was already shown that the T4SS can be used to deliver other nucleases such as Sce1 and ZFNs and recombination proteins such as Cre/lox into plants.

Our response: Thanks for bringing these points to our attention. We added these points to the discussion.

Reviewer #2 (Remarks to the Author):

Improvement of the plant transformation efficiency in recalcitrant plant species/varieties is a major issue in crop improvement applications by cis/transgenic and CRISPR technologies. This manuscript describes a work combining two known concepts: (1) Presence of type III effectors (T3Es), such as AvrPto, in the plant cell can suppress plant immune response and improve the efficiency of Agrobacterium-mediated transformation (AMT); (2) The *Pseudomonas syringae* type III secretion system (T3SS) can be functionally expressed in other Gram-negative bacteria. Since the previous applications of (1) required transgenic expression of T3E in the plant, it had a limitation in applications. The idea of combining these two in this manuscript is a good one, and thus the impact of the work mainly depends on how well this idea worked out in practice. In this light, the key results are in Fig 4 – AMT of the wheat cultivar Fielder. On average 4-fold efficiency improvement by AvrPto sounds good.

However, Fielder is not AMT-recalcitrant among wheat varieties. According to the authors' discussion, this improvement is based on a transformation method, which is not a very good method to start with. Thus, the improved AMT efficiency achieved in this study is not as high as the AMT efficiency achieved by refinements of the method (ref 36). It is not known if the method described in this manuscript could further improve the efficiency from the level achieved by the refined method.

Our response: Recalcitrant to transformation could be due to several factors. The two main factors are **plant defense responses** and the **inability of the plant explant to regenerate**. The later has been addressed by overexpressing plant development regulators such as *LEAFY COTYLEDON1*, *LEAFY COTYLEDON2*, *WUSCHEL (WUS)*, *BABY BOOM (BBM)*, *GROWTH-REGULATING FACTOR 4 (GRF4)* and its cofactor *GRF-INTERACTING FACTOR 1 (GIF1)*. Our aim in this manuscript is to overcome plant defense

responses and not to improve plant regeneration. We may not be able to increase transformation efficiency by plant defense suppression in a recalcitrant wheat variety that is difficult to regenerate. However, the strategy described in the manuscript can be potentially used in the future to deliver WUS, BBM, or GRF proteins to increase plant regeneration. In our opinion, using a refined method of transformation will not change the fold difference observed in transformation efficiency in the engineered *Agrobacterium*. In our transformation method, we incorporated all the optimized conditions like stage of donor material and pre-treatment by centrifugation. Most of the steps in that refined paper are included in our protocol. The transformation efficiencies in both the protocols (the refined and ours) are same.

Furthermore, the authors should have tried with truly AMT-recalcitrant varieties of wheat – if the method works with elite varieties with no transformation success records, it would have an impact for general audience.

Our response: This is a good suggestion. However, most varieties are recalcitrant to transformation because they are difficult to regenerate. The main aim of this paper is not to improve plant regeneration but to improve *Agrobacterium*-mediated plant transformation (delivery and integration of T-DNA) by dampening plant defense responses. As explained above, our strategy can be used in the future to enhance plant regeneration.

Improvement of the AMT efficiency in *Arabidopsis* and *N. benthamiana* has little impact for the practical purpose: nobody uses tissue-culture-based AMT for *Arabidopsis* as the floral dip AMT method is highly efficient and much less cumbersome; *N. benthamiana* has a high efficiency to start with. Also, the improvement of the transformation efficiency is rather moderate (all less than 2-fold, except for transient transformation). Thus, the results in *Arabidopsis* and *N. benthamiana* should be considered only for the method characterization.

Our response: We agree that *Arabidopsis* and *N. benthamiana* results have less impact for practical purpose. We used these to show the proof of concept and to show the effect in more than one plant species. The reason we didn't see a big difference in transformation efficiency is because these plants are already highly susceptible to *Agrobacterium* infection. The reviewer comments have now made us curious if the engineered *Agrobacterium* can enhance floral dip transformation in *Arabidopsis*. Results obtained from our assay is presented as Supplementary Figure 4.

Suppression of immune response by pathogen effectors is the matter of timing and strength. It is likely that delivery of T3E to the plant cell is not sufficiently fast and/or not sufficiently abundant to have a strong suppression effect. This limited level of immune response suppression is clear from Fig 5.

Our response: We agree with the reviewer that there is only limited immune response suppression. We feel this limited level of immune response suppression is enough to make the plant more susceptible to *Agrobacterium* infection. When *Pseudomonas syringae* infects, it delivers ~ 30 type III effectors and make the plant more susceptible to pathogen infection. The amount of effector protein delivered from *Pseudomonas* and engineered *Agrobacterium* should be similar. In our strategy, we are delivering only

one effector at a time that gives only subtle difference in the immune suppression and fortunately, it is sufficient to increase plant's susceptibility to *Agrobacterium* infection.

In summary, I do not think that this work has demonstrated a sufficient impact in AMT improvement for general audience. This manuscript better fits to a more specialized journal for plant transformation.

Our response: We understand the reviewer's concern. We have now carried out transformation assays in few more crop plants such as alfalfa and switchgrass (Results are presented in Figures 4 and 6). In our opinion this is a novel strategy to improve plant transformation and also has the potential to deliver other proteins into plants in a non-transgenic manner. We therefore believe that the manuscript is suitable of general audience.

Specific points.

1. Page 2, abstract and other places. "deliver the *P. syringae* effectors AvrPto, AvrPtoB, and HopAO1" This expression gives an impression that all three effectors were delivered at once. It should be clear that it was delivered one by one but not in combination.

Our response: We changed this line to "individually deliver the *P. syringae* effectors AvrPto, AvrPtoB, or HopAO1"

2. Page 4, middle. "practically this strategy cannot be used in the field because these transgenic plants will be susceptible to other pathogens". This is not a good argument because the transgene can be segregated out. The real limitation of the approach is that the plants have to be already transformed with T3E transgenes.

Our response: We agree with reviewer's suggestion and modified this line.

3. Page 6, middle. "In addition to demonstrating that the T3SS is functional in *A. tumefaciens* to secrete T3E in culture and translocate them to plant cells, assembly of a full-length GFP by independent translocation through T4SS and T3SS highlights cooperation of both systems in inter-kingdom T-DNA transfer with T3SS translocating proteins and T4SS translocating both proteins and T-DNA." This sentence is not clear. Translocated proteins are different for T3SS and T4SS. And the proteins translocated by T4SS are not engineered proteins.

Our response: We have modified the sentence as we mentioned in our reply to reviewer 1.

4. Page 7, top. The floral dip method is the commonly used method in *Arabidopsis* in practice. For a practical purpose, the transformation efficiency by the floral dip method should be investigated.

Our response: We thank the reviewer for this suggestion. Floral dip transformation was done to compare non-tumorigenic *Agrobacterium* strains expressing AvrPto with and without T3SS. Negative control strains were also included. Surprisingly, we saw greater than two-fold increase in transformation efficiency by the engineered *Agrobacterium* as shown in Supplemental Figure 4.

5. Page 7, middle, Fig 2gh. Is the fresh weight of leaf disks a good measure of the weight for the tumor tissue? Why not count the number of independent transgenic plants, like Fig 4?

Our response: Generating transgenic plants as a measure of stable transformation process is a time consuming one. There are many reports (Vaghchhipawala et al., 2012; Anand et al., 2008; Anand et al., 2007; Wang et al., 2018) using fresh weight of leaf disks for the tumor assay. Our goal here is to show the stable transformation efficiency and, in our opinion, generating stable lines is not required. We generated transgenic lines in the crop plants.

6. Fig 4. Instead of showing % of independent transgenic plants, it should show the number of independent transgenic plants (per certain number of plates used for the tissue culture procedure).

Our response: We included another supplementary figure (Supplementary Figure 8) that show the number of independent transgenic plants.

Also, the AGL1(pANIC6B, T3E, pLN18) must be compare to AGL1(pANIC6B) in the posthoc tests. In addition, one-way ANOVA assumes equal variance. Since the variance for high values are much larger, one-way ANOVA cannot be used directly – the authors can use Bartlett test for equal variance across the strains. Probably log-transformation works for equal variance.

Our response: We thank the reviewer for the suggestion. We revised our graph to reflect comparison of AGL1 (pANIC6B, T3E, pLN18) with AGL1 (pANIC6B). We combined the results from two experiments and run the statistics. Variance test result is significant in one-way ANOVA. Therefore, we did two-way ANOVA followed by Tukey's test.

7. Page 10, bottom. “we also observed a greater fluorescent signal using GFP compared to PhiLOV.” Where is the data? It must be quantitative data as it is a quantitative statement.

Our response: We didn't quantify the fluorescence. Since this data is not crucial, we removed this line from the manuscript.

8. Page 11, top. “these effectors target plant immune signaling components through effector-triggered immunity (ETI) and not PAMP receptors.” This statement is wrong. There are reports that AvrRpt2 and AvrRpm1 compromise basal immunity (i.e., non-ETI).

Our response: Here we wanted to highlight that the effectors we chose target very early stage of PTI pathway at the PAMP receptor level. Therefore, we re-worded this sentence.

Reviewer #3 (Remarks to the Author):

The authors have used *Agrobacterium tumefaciens* to express a type III secretion system (T3SS) from *Pseudomonas syringae* and deliver the *P. syringae* effectors AvrPto, AvrPtoB, and HopAO1 to suppress host defense responses. They also report that engineered *A. tumefaciens* expressing a T3SS can deliver histone H2A-1 protein to enhance AMT.

Using T3SS to deliver proteins into plant cells to enhance *Agrobacterium*-mediated transformation efficiencies is innovative.

However, for AvrPto, AvrPtoB, and HopAO1, they authors used the three plant species, *Arabidopsis*, *N. benthamiana* and wheat for the experiments, However, the increases in transformation efficiency of both *Arabidopsis* and *N. benthamiana* are not in significant while about 400% increase was observed for wheat. However, if only one of the three plant species tested shows significant improvement in transformation efficiency, the question is how widely applicable is their technology for other plant species.

Regarding the use of an engineered *A. tumefaciens* expressing the T3SS to deliver the histone H2A-1 protein into plant cell to enhance AMT. *Arabidopsis* and *N. benthamiana* were used for the experiment.

Our Response: We have added *HTA1* data for wheat to Figure 6d. In addition to this we have now added transformation data to two additional crop species, alfalfa and switchgrass as Figures 4b-c and 6e-f.

Again, the increases in transformation efficiencies of the two plant species are insignificant.

Our response: The difference in transformation efficiency of *Arabidopsis* and *N. benthamiana* with engineered *Agrobacterium* strain was statistically significant. We agree that the fold difference is not very dramatic. This is not surprising since both *Arabidopsis* and *N. benthamiana* are already very susceptible to *Agrobacterium* transformation. We therefore used wheat that is a bit recalcitrant to transformation. Now we have data from transformation assays in two other crop species as well.

There are also some other minor problems.

Overall, I feel that the increases in transformation efficiency presented in the manuscript are not significant enough. Also, the applicability of the described technologies in other plant species is uncertain.

Our response: We feel that ~400% increase in transformation of three different crop species is very significant. By including two more crop species, we showed broader applicability of our technology.

Reviewers' Comments:

Reviewer #1:

Remarks to the Author:

The ms by Raman et al describes the introduction of a cosmid encoding a Pseudomonas T3SS in Agrobacterium for delivery of effector proteins in plant host cells. The results show that this T3SS can transfer proteins from Agrobacterium to plants and that this can be used for enhancing Agrobacterium-mediated transformation. In the revised ms this latter point is expanded by the inclusion of alfalfa and switchgrass for transformation.

I have only few comments:

line 124/125 the authors have reformulated this sentence and mention now: "highlights that both T3SS and T4SS can operate in the same bacterium" In fact they use two different strains in this experiment, one with the pTi vir T4SS and the other with the T3SS. The authors mean to say, I guess, that in the bacterial species Agrobacterium both a T4SS and a T3SS can operate. This in itself is not unusual. Many bacteria employ a T4SS for conjugative transfer, and these same bacteria may have a T3SS, which may be present in the same cell and even on the same plasmid. Other bacteria, such as Rhizobium loti employ either a T3SS or a T4SS for protein delivery to plant host cells during nodulation. Therefore I do not think what is said in line 124/125 has any significance.

(By the way the strains used in this experiment are missing in the table of strains: which helper was used with GFP1-10?).

line 287 polyploid

line 324 the potential of delivery of Cas proteins by T3SS is mentioned as well as the previous results seen with Cas delivery by T4SS. The authors mention that delivery by T4SS is inefficient, suggesting that they expect that delivery by the T3SS will be more efficient. However, this remains to be seen. There are no quantitative data available about the amounts of proteins delivered into host cells by either the T3SS or the T4SS.

line 631 correct reference, which is still in reference manger format

Fig4 Although I appreciate the increase in transformation efficiency seen in the floral dip experiment, I found the transformation frequencies observed extremely low. In my experience floral dip generates transformation with frequencies of about 1%. The authors see only 1:4000 and an increase to 16:4000, which is still way lower than what we see.

Reviewer #2:

Remarks to the Author:

My main concern of the work was whether the technology gives substantial improvement in applications to crop species. This is the authors' response to my concern on wheat transformation: "... most (wheat) varieties are recalcitrant to transformation because they are difficult to regenerate. The main aim of this paper is not to improve plant regeneration..." This is a disappointing fact in the context of this work. The authors must have in the main text a statement, "Since plant regeneration is the bottleneck in generation of transgenic wheats in most cases, the impact of this work on transgenic wheat generation is limited." to properly inform general audience.

I am also concerned about the measure of the transformation efficiency, "% of independent transgenic plants/shoots". Is this really the right measure of the transformation efficiency? It cannot exclude the possibility that the decreased number resulted in a high percentage. What about the number of independent transgenic plants/shoots per experiment? At least the authors must make the raw count data available as supplements.

Reviewer #3:

Remarks to the Author:

The authors report a strategy that can enhance efficiencies of Agrobacterium mediated plant transformation up to 400%. They engineered Agrobacterium tumefaciens to express a type III secretion system (T3SS) from Pseudomonas syringae, and individually deliver the P. syringae effectors AvrPto, AvrPtoB, or HopAO1 to suppress host defense responses. It appears that the method works well in more than one plant species. Comparing to chemical manipulations of Agrobacterium cells, or explants before, during and/or after infection (i.e., Agro and explant co-incubation), however, this method is much more complex to use. Some simple chemical treatment can also lead several fold increase in plant transformation efficiency.

The authors have also showed that engineered A. tumefaciens expressing a T3SS can help deliver a plant protein, histone H2A-1, into the infected plant cells to enhance transformation efficiency. This can be significant for enhancing both transient expression of T-DNA genes and stable transformation efficiency of recalcitrant plant species. The engineered Agrobacterium may also be a useful vehicle to deliver other proteins into plant cells such as Cas9 for a non-transgenic gene editing.

I feel that the authors address my concerns well. Overall, the approaches presented are novel and the increase in transformation efficiency is significant. However, it is not hard to predict how easily and widely these methods will be used by others who are doing plant genetic transformation or gene editing.

Reviewer #1 (Remarks to the Author):

The ms by Raman et al describes the introduction of a cosmid encoding a *Pseudomonas* T3SS in *Agrobacterium* for delivery of effector proteins in plant host cells. The results show that this T3SS can transfer proteins from *Agrobacterium* to plants and that this can be used for enhancing *Agrobacterium*-mediated transformation. In the revised ms this latter point is expanded by the inclusion of alfalfa and switchgrass for transformation.

I have only few comments:

line 124/125 the authors have reformulated this sentence and mention now: "highlights that both T3SS and T4SS can operate in the same bacterium" In fact they use two different strains in this experiment, one with the pTi vir T4SS and the other with the T3SS. The authors mean to say, I guess, that in the bacterial species *Agrobacterium* both a T4SS and a T3SS can operate. This in itself is not unusual. Many bacteria employ a T4SS for conjugative transfer, and these same bacteria may have a T3SS, which may be present in the same cell and even on the same plasmid. Other bacteria, such as *Rhizobium loti* employ either a T3SS or a T4SS for protein delivery to plant host cells during nodulation. Therefore I do not think what is said in line 124/125 has any significance.

Our response: Here we wanted to highlight that both T3SS and T4SS that is involved in inter-kingdom DNA transfer can occur in the same bacterium. We agree with the reviewer that T3SS and T4SS are from different strain in confocal microscopy experiment whereas both T3SS and T4SS are present in some of the strains used for *Arabidopsis* transient assay experiments for example in EHA105 (pCAMBIA1301, *AvrPto*, pLN18), *AvrPto* is delivered by T3SS and β -glucuronidase expressed from T-DNA delivered by T4SS. Therefore, we have now removed "the same bacterium" from the sentence. To our knowledge, we are not aware of any reports that show both T3SS and T4SS can operate in the same *Agrobacterium*.

(By the way the strains used in this experiment are missing in the table of strains: which helper was used with GFP1-10?).

Our response: Thanks for bringing this point to our attention. We have now included strain information in row 36 of Supplementary table 1.

line 287 polyploid

Our response: We removed polyploid

line 324 the potential of delivery of Cas proteins by T3SS is mentioned as well as the previous results seen with Cas delivery by T4SS. The authors mention that delivery by T4SS is inefficient, suggesting that they expect that delivery by the T3SS will be more efficient. However, this remains to be seen. There are no quantitative data available about the amounts of proteins delivered into host cells by either the T3SS or the T4SS.

Our response: We understand the reviewer's concern. We removed this line.

line 631 correct reference, which is still in reference manger format

Our response: Corrected.

Fig4 Although I appreciate the increase in transformation efficiency seen in the floral dip experiment, I found the transformation frequencies observed extremely low. In my experience floral dip generates transformation with frequencies of about 1%. The authors see only 1:4000 and an increase to 16: 4000, which is still way lower than what we see.

Our response: Because of high transformation efficiency of floral dip transformation method, it is hard to see any subtle differences between the treatments when normal (high) *Agrobacterium* concentration is used. Therefore, we reduced the *Agrobacterium* concentration for our floral dip transformation method to see subtle differences between the *Agrobacterium* strains used. This is the reason we see an overall reduction in transformation efficiency. We have now mentioned the same in the results section so that the reader can understand the low transformation frequency observed.

Reviewer #2 (Remarks to the Author):

My main concern of the work was whether the technology gives substantial improvement in applications to crop species. This is the authors' response to my concern on wheat transformation: "... most (wheat) varieties are recalcitrant to transformation because they are difficult to regenerate. The main aim of this paper is not to improve plant regeneration..." This is a disappointing fact in the context of this work. The authors must have in the main text a statement, "Since plant regeneration is the bottleneck in generation of transgenic wheats in most cases, the impact of this work on transgenic wheat generation is limited." to properly inform general audience.

Our response: We thank the reviewer for the suggestion – we now included few lines to reflect the reviewer's suggestion.

Transformation and regeneration are major bottlenecks in generation of transgenic plants or genome edited plants. Our study aimed to improve transformation at the delivery and integration of T-DNA steps. Chimeric protein of plant developmental gene encoding GROWTH-REGULATING FACTOR 4 (GRF4) and its cofactor GRF-INTERACTING FACTOR 1 (GIF1) (GRF4-GIF1) and GRF5 were shown to enhance regeneration and transformation of both monocot and dicot species (Debernardi et al., 2020; Kong et al., 2020). Specific morphogenic genes that are known to induce somatic embryogenesis and regeneration are used to improve transformation efficiency in monocots (Lowe et al. 2016; Lowe et al. 2018). Since constitutive expression of these genes have negative phenotypic and reproductive effects, altruistic transformation is used in maize and sorghum which uses transient expression of *Baby boom* and *Wuschel2* to promote somatic embryogenesis and regeneration in nearby transformed cells (Hoerster et al. 2020; Aregawi et al. 2021). In the future, these proteins can be potentially delivered through *Agrobacterium* strain expressing T3SS to improve somatic embryogenesis and plant regeneration.

I am also concerned about the measure of the transformation efficiency, "% of independent transgenic plants/shoots". Is this really the right measure of the transformation efficiency? It cannot exclude the possibility that the decreased number resulted in a high percentage. What about the number of independent transgenic plants/shoots per experiment? At least the authors must make the raw count data available as supplements.

Our response: We thank the reviewer for the suggestion. Raw data count is available in source data.

Reviewer #3 (Remarks to the Author):

The authors report a strategy that can enhance efficiencies of *Agrobacterium* mediated plant transformation up to 400%. They engineered *Agrobacterium tumefaciens* to express a type III secretion system (T3SS) from *Pseudomonas syringae*, and individually deliver the *P. syringae* effectors AvrPto, AvrPtoB, or HopAO1 to suppress host defense responses. It appears that the method works well in more than one plant species. Comparing to chemical manipulations of *Agrobacterium* cells, or explants before, during and/or after infection (i.e., Agro and explant co-incubation), however, this method is much more complex to use. Some simple chemical treatment can also lead several fold increase in plant transformation efficiency.

The authors have also showed that engineered *A. tumefaciens* expressing a T3SS can help deliver a plant protein, histone H2A-1, into the infected plant cells to enhance transformation efficiency. This can be significant for enhancing both transient expression of T-DNA genes and stable transformation efficiency of recalcitrant plant species. The engineered *Agrobacterium* may also be a useful vehicle to deliver other proteins into plant cells such as Cas9 for a non-transgenic gene editing.

I feel that the authors address my concerns well. Overall, the approaches presented are novel and the increase in transformation efficiency is significant. However, it is not hard to predict how easily and widely these methods will be used by others who are doing plant genetic transformation or gene editing.

Our response: We thank the reviewer for the appreciation and support in publishing this work. We anticipate many people will use our engineered *Agrobacterium* strain to improve transformation efficiency of recalcitrant plant species. We are already working with a company who is using our engineered *Agrobacterium* strain to improve plant transformation efficiency of recalcitrant crop species.

Reviewers' Comments:

Reviewer #1:

Remarks to the Author:

The ms by Raman et al describes the introduction of a cosmid encoding a Pseudomonas T3SS in Agrobacterium for delivery of effector proteins in plant host cells. The results show that this T3SS can indeed transfer proteins from Agrobacterium to plants and that this can be used for enhancing Agrobacterium-mediated transformation, at least under certain conditions.

The authors have addressed the remaining points which I raised in the revised ms.

Reviewer #2:

Remarks to the Author:

In my first review, I proposed to apply the technology the authors developed to transformation of a truly recalcitrant wheat variety to demonstrate true utility of the technology. The authors refused to perform the experiment, saying that the regeneration efficiency is the bottleneck rather than the transformation efficiency in such wheat varieties. Thus, in my second review, I demanded to state it in the manuscript as it is very important information to readers. Now the authors changed to say both the regeneration and transformation efficiencies are important. It appears that the authors lied to avoid an additional experiment in response to my first review.

Response to reviewer comments

REVIEWERS' COMMENTS

Reviewer #1 (Remarks to the Author):

The ms by Raman et al describes the introduction of a cosmid encoding a Pseudomonas T3SS in Agrobacterium for delivery of effector proteins in plant host cells. The results show that this T3SS can indeed transfer proteins from Agrobacterium to plants and that this can be used for enhancing Agrobacterium-mediated transformation, at least under certain conditions. The authors have addressed the remaining points which I raised in the revised ms.

Our Response: We thank the reviewer for accepting our manuscript.

Reviewer #2 (Remarks to the Author):

In my first review, I proposed to apply the technology the authors developed to transformation of a truly recalcitrant wheat variety to demonstrate true utility of the technology. The authors refused to perform the experiment, saying that the regeneration efficiency is the bottleneck rather than the transformation efficiency in such wheat varieties. Thus, in my second review, I demanded to state it in the manuscript as it is very important information to readers. Now the authors changed to say both the regeneration and transformation efficiencies are important. It appears that the authors lied to avoid an additional experiment in response to my first review.

Our Response: Both regeneration and transformation efficiencies are important to successfully transform a plant. Most wheat varieties cannot be regenerated and therefore cannot be transformed. We didn't really lie about this. Perhaps the reviewer wants us to mention about the drawback our technology where in it cannot improve regeneration. We have now added the below sentence in the discussion section.

“The limitation of this strategy is that it cannot be used in plant varieties that are recalcitrant to regeneration.”